# Reliability and validity of the UK Biobank cognitive tests

Chloe Fawns-Ritchie[1,2]*, Ian J. Deary[1,2]

**1** Centre for Cognitive Ageing and Cognitive Epidemiology, University of Edinburgh, Edinburgh, United Kingdom, **2** Department of Psychology, University of Edinburgh, Edinburgh, United Kingdom

* c.fawns-ritchie@ed.ac.uk

## Abstract

UK Biobank is a health resource with data from over 500,000 adults. The cognitive assessment in UK Biobank is brief and bespoke, and is administered without supervision on a touchscreen computer. Psychometric information on the UK Biobank cognitive tests are limited. Despite the non-standard nature of these tests and the limited psychometric information, the UK Biobank cognitive data have been used in numerous scientific publications. The present study examined the validity and short-term test-retest reliability of the UK Biobank cognitive tests. A sample of 160 participants (mean age = 62.59, SD = 10.24) was recruited who completed the UK Biobank cognitive assessment and a range of well-validated cognitive tests ('reference tests'). Fifty-two participants returned 4 weeks later to repeat the UK Biobank tests. Correlations were calculated between UK Biobank tests and reference tests. Two measures of general cognitive ability were created by entering scores on the UK Biobank cognitive tests, and scores on the reference tests, respectively, into separate principal component analyses and saving scores on the first principal component. Four-week test-retest correlations were calculated for UK Biobank tests. UK Biobank cognitive tests showed a range of correlations with their respective reference tests, i.e. those tests that are thought to assess the same underlying cognitive ability (mean Pearson $r$ = 0.53, range = 0.22 to 0.83, $p \leq$ .005). The measure of general cognitive ability based on the UK Biobank cognitive tests correlated at $r$ = 0.83 (p < .001) with a measure of general cognitive ability created using the reference tests. Four-week test-retest reliability of the UK Biobank tests were moderate-to-high (mean Pearson $r$ = 0.55, range = 0.40 to 0.89, $p \leq$ .003). Despite the brief, non-standard nature of the UK Biobank cognitive tests, some tests showed substantial concurrent validity and test-retest reliability. These psychometric results provide currently-lacking information on the validity of the UK Biobank cognitive tests.

## Introduction

UK Biobank is a large prospective cohort study that was designed to investigate the health of middle-aged and older adults residing in the UK (https://www.ukbiobank.ac.uk/) [1]. At baseline (2006–2010), over half a million participants aged 40 to 70 years attended a UK Biobank clinic and completed a touchscreen questionnaire collecting information on health and lifestyle. Physical measurements and biological samples were also collected during this clinic visit. Subsamples of UK Biobank participants have undergone repeat testing. Between 2009 and

**Funding:** Join Dementia Research was used to identify potential participants for this study. Join Dementia Research is funded by the Department of Health and delivered by the National Institute for

Health Research in partnership with Alzheimer
Scotland, Alzheimer's Research UK and
Alzheimer's Society. This work was supported by
the University of Edinburgh Centre for Cognitive
Ageing and Cognitive Epidemiology, part of the
cross council Lifelong Health and Wellbeing
Initiative, funded by the Biotechnology and
Biological Sciences Research Council (BBSRC),
and Medical Research Council (MRC) (grant
number MR/K026992/1). CF-R and IJD were
supported by DPUK, funded through the MRC
(MR/L023784/2). The funders had no role in study
design, data collection and analysis, decision to
publish, or preparation of the manuscript.

**Competing interests:** The authors have declared
that no competing interests exist.

2013, 20,000 UK Biobank participants returned and completed the baseline assessment again. Currently, UK Biobank are conducting an imaging study (started in 2014) in which 100,000 UK Biobank participants are being asked to complete the baseline assessment again and undergo brain and body scanning. In addition to these clinic-based assessments, subsamples have completed web-based assessments, including web-based cognitive testing. More information on the data collected at each assessment is reported elsewhere [1].

A cognitive assessment was administered as part of the fully-automated touchscreen questionnaire [1]. The original UK Biobank cognitive assessment was very brief (approximately 5 minutes). At baseline, almost all participants completed the Pairs Memory test, a test of visual memory, and the Reaction Time test, a measure of processing speed. Subsamples also completed tests of working memory (Numeric Memory test), prospective memory (Prospective Memory), and verbal and numerical reasoning (Fluid Intelligence) [1]. This test battery, excluding Numeric Memory, was then administered again at the repeat visit that re-tested 20,000 people. Subsequently, more detailed measures of cognitive function assessing cognitive domains known to decline with increasing age [2] were introduced at the imaging study. The new tests included tests of verbal declarative memory, executive function, and non-verbal reasoning. To estimate prior cognitive functioning, a measure of vocabulary was also included, which tends to remain relatively stable across the age range assessed in UK Biobank [2]. More information on the cognitive tests and domains assessed at each UK Biobank assessment are reported in S1 File (S1 Table).

Cognitive functions are typically assessed using well-validated cognitive tests administered under standardised conditions by a trained psychological tester, which may be seen as the gold standard method of cognitive assessment. Probably owing to the size of the UK Biobank sample, and the magnitude of other data being collected, the method of cognitive data collection was made efficient via a brief and fully-automated touchscreen assessment that was administered unsupervised [1]. That is, no tester observed the UK Biobank participants while they completed the cognitive assessment. Some of the UK Biobank cognitive tests were specifically designed for use in UK Biobank, whereas others were adapted versions of commonly-used tests which have been modified for use at the fully-automated touchscreen assessment.

The cognitive testing section of the UK Biobank data showcase (http://biobank.ndph.ox.ac.uk/showcase/label.cgi?id=100026) provides a brief description of each of the cognitive tests, and some basic descriptive statistics (e.g., distribution of test scores, mean and standard deviation); however, there is little other information regarding why these specific tests were chosen or how these tests were developed. UK Biobank did not examine the concurrent validity and test-retest reliability of these cognitive tests. Despite this, many research papers have been published using the UK Biobank cognitive test data (https://www.ukbiobank.ac.uk/published-papers/). The number of studies using the UK Biobank cognitive data will likely increase now that UK Biobank have released genotyping data on all 500,000 participants, and brain imaging data is available on a subsample of UK Biobank participants [1].

Before researchers can have confidence in the quality of the UK Biobank cognitive test data it is important that the psychometric properties of these tests are investigated. Using the UK Biobank data collected at baseline and the repeat visit, one study [3] examined the stability (test-retest reliability) of the original UK Biobank tests over a four year interval. Whereas this study found that Fluid Intelligence (correlation between baseline and repeat $r = 0.65$) and Reaction Time ($r = 0.54$) showed substantial stability [3], the test-retest reliability of Pairs Matching was low ($r = 0.19$) [3]. Scores on reliable cognitive tests should be relatively stable over a short interval. Typical test-retest intervals are usually between 2–4 weeks. The test-retest interval in Lyall et al. [3] was much longer than typical test-retest intervals, and could have included ageing effects. Therefore, this might not be a useful indicator of the shorter-term

stability of these tests. No one has examined the test-retest reliability of the newer UK Biobank tests introduced since baseline.

One of the most replicated findings in psychological research is that performance on tests of cognitive function are positively correlated and that a measure of general cognitive ability (*g*) can be extracted from scores on a diverse set of cognitive tests [4–7]. One way to examine the validity of the UK Biobank cognitive assessment would be to confirm that these brief and unsupervised cognitive tests are positively correlated. Principal component analysis (PCA) is often used to examine the inter-correlational structure of cognitive tests. A composite score, which is created by saving scores based on the first unrotated principal component, typically accounts for about 40% of the variance in a wide range of different cognitive tests [8, 9]. Using UK Biobank baseline data, Lyall et al. [3] entered scores on the UK Biobank Reaction Time, Pairs Matching, Fluid Intelligence and Numeric Memory tests into a PCA model. The first unrotated principal component accounted for 40% of the variance and the individual test scores all loaded at $\geq 0.49$ on this component [3], confirming the scores were positively correlated and that a *g* component was present in the original UK Biobank tests. The correlational structure of the newer UK Biobank tests introduced since baseline have not been investigated.

The aim of the current study was to expand on the work carried out by Lyall et al. [3] and investigate aspects of the reliability and validity of the UK Biobank cognitive tests; both the original baseline tests and newer tests introduced since baseline. In the current study, we recruited an independent sample of participants who had not taken part in the UK Biobank study. These participants were administered the enhanced UK Biobank cognitive assessment that was given during the imaging study and included all baseline tests as well as more detailed tests introduced since baseline (S1 File, S1 Table). Participants also completed a battery of well-validated, standard cognitive tests; hereinafter we will call these 'reference tests'. For each of the UK Biobank tests, we chose a reference test that we judged was assessing the same underlying cognitive domain. Participants also completed brief screening tests of cognitive impairment and measures of subjective memory complaints, which are often applied in studies of normal and pathological ageing to measure global cognitive functioning (hereinafter we will call these 'general' tests). Approximately four weeks after the baseline assessment, a subsample of participants returned and repeated the enhanced UK Biobank cognitive assessment.

To investigate the reliability and validity of the UK Biobank tests, three sets of analyses were carried out. First, the concurrent validity of the UK Biobank tests was investigated by correlating scores on the UK Biobank tests with scores on the reference tests. We predicted that the correlations between UK Biobank tests and reference tests that assessed a similar cognitive domain should show higher correlations than those between UK Biobank tests and reference tests that assessed different cognitive domains. Second, we investigated whether a *g* component was present in the correlations among the unsupervised UK Biobank cognitive tests, and tested whether any such *g* component correlated highly with a *g* component created using the reference tests which were administered by a trained tester under standardised conditions. The present study predicted that the correlation between a *g* component created using the UK Biobank tests and a *g* component created using the reference tests would be high (e.g., $r > 0.7$). Third, we characterised the short-term test-retest stability of the UK Biobank tests by correlating the cognitive test scores from time 1 and time 2 (approximately 4 weeks apart).

## Materials and methods

### Participants

A sample of participants who had not taken part in the UK Biobank study was recruited. Participants were identified through the University of Edinburgh Volunteer Panel and Join

Dementia Research (https://www.joindementiaresearch.nihr.ac.uk/home?login). Both are databases of volunteers who are interested in taking part in research. Potential participants who were aged 40 to 80 years old and who were able to travel to the Psychology Department at the University of Edinburgh were contacted and invited to take part in this study. The age range of 40 to 80 years was used in the current study because this is approximately the age range of UK Biobank participants across the various data collection points to date (http://biobank.ndph.ox.ac.uk/showcase/field.cgi?id=21003). The UK Biobank baseline study aimed to recruit participants aged 40 to 70 years (mean age = 56.53, SD = 8.10). These participants have become older as they have been followed up. The age range of UK Biobank participants at the UK Biobank imaging study—which uses the same cognitive assessment as in the current study—was 44 to 82 years (mean age = 62.59 years, SD = 10.24).

UK Biobank participants and people with a diagnosis of dementia or mild cognitive impairment were not eligible for this study. A total of 160 participants were recruited. Written informed consent was obtained for all participants. This study received ethical approval from the University of Edinburgh Psychology Research Ethics Committee (reference number 2-1718/3).

## Materials

**UK Biobank cognitive test battery.**   UK Biobank provided us with a stand-alone version of the UK Biobank cognitive test battery that was administered at the UK Biobank imaging study. To make the testing session as similar as possible to the UK Biobank clinic study sessions, the present study used the same touch screen monitor and computer setup as that which is used at the UK Biobank clinics' imaging sessions. The UK Biobank cognitive assessment was designed to be fully-automated and the tests were administered unsupervised. During the UK Biobank clinic assessment, there are UK Biobank staff present in the clinic while participants are completing the cognitive assessment; however, participants are expected to sit and work through the cognitive assessment independently. All instructions for the UK Biobank cognitive tests are presented onscreen. In the present study, participants were given brief oral instructions that they were going to complete some tasks on the computer on their own, and that they were to follow the instructions on the screen. To emulate the UK Biobank testing conditions, one author (CF-R) was present in the room while participants completed the UK Biobank tests, but participants were left to work through the tests on their own. The tests administered as part of the UK Biobank cognitive assessment are listed in Table 1 and a detailed description of each test's contents, administration, and scoring is provided in the S1 File.

**General tests.**   A cognitive screening test and a subjective memory questionnaire (see Table 1) were included in the current study to investigate the correlations between the UK Biobank cognitive tests and frequently-used measures of global cognitive function. Detailed descriptions of these tests are provided in S1 File.

**Reference tests.**   To be able to test the concurrent validity of each UK Biobank test, a battery of standard neuropsychological tests was administered. For each UK Biobank test, we selected one or more well-validated, standard cognitive tests that resembles the UK Biobank test in terms of the underlying cognitive domain thought to be assessed, and in the actual content of the task ('reference tests'). The reference tests were all administered under standardised conditions, one-to-one, face-to-face, by a trained tester, strictly following the administration instructions. The reference tests chosen for each UK Biobank test are shown in Table 1. A detailed description of each of the reference tests' contents, administration, and scoring is provided in S1 File. No reference test was chosen for the UKB Fluid IQ test because no test was identified by the authors as a suitable comparator.

**Table 1. UK Biobank cognitive tests, general cognitive tests, and reference cognitive tests administered in the current study.**

| Cognitive domain | UK Biobank tests | | General and reference tests | |
| --- | --- | --- | --- | --- |
| | Test name | Abbreviation | Reference test and source | Abbreviation |
| Visual declarative memory | Pairs Matching Test | UKB Pairs Matching | Wechsler Memory Scale IV Designs I and Designs II [24] | WMS-IV Designs I WMS-IV Designs II |
| Processing speed | Reaction Time Test | UKB RT | Deary-Liewald Reaction Time Test Simple Reaction Time and Choice Reaction Time [17] | DLRT Simple RT DLRT Choice RT |
| Prospective memory | Prospective Memory Test | UKB Prospective Memory | Rivermead Behavioural Memory Test—Extended Version Appointments [25] | RMBM Appointments |
| Verbal and numerical reasoning | Fluid Intelligence Test | UKB Fluid IQ | - | - |
| Working Memory | Numeric Memory Test | UKB Numeric Memory | Wechsler Adult Intelligence Scale IV Digit Span Forwards, Backwards and Sequence [26] | WAIS-IV Digit Span Forwards WAIS-IV Digit Span Backwards WAIS-IV Digit Span Sequence |
| Executive function | Trail Making Test parts A and B[a] | UKB TMT part A UKB TMT part B | Trail Making Test parts A and B [16] | TMT part A TMT part B |
| Processing speed | Symbol Digit Substitution Test | UKB Symbol Digit | Symbol Digit Modalities Test [11] | SDMT |
| Crystallised ability | Picture Vocabulary[b] | UKB Picture Vocabulary | Peabody Picture Vocabulary Test, Fourth Edition [21] | PPVT |
| | | | National Adult Reading Test [27] | NART |
| | | | NIH Toolbox Picture Vocabulary Test [13] | NIH Toolbox Picture Vocabulary |
| Verbal declarative memory | Paired Associate Learning Test[c] | UKB PAL | Wechsler Memory Scale IV Verbal Paired Associates I and Verbal Paired Associates II [24] | WMS-IV VPA I WMS-IV VPA II |
| Executive function | Tower Rearranging Test[d] | UKB Tower Test | Delis-Kaplan Executive Function System Tower Test [28] | D-KEFS Tower Test |
| Non-verbal reasoning | Matrix Pattern Completion[e] | UKB Matrices | COGNITO Matrices [23] | COGNITO Matrices |
| Subjective memory | - | - | Self-rated memory questions [29] | Self-rated memory |
| Global cognitive function | - | - | Mini Addenbrooke's Cognitive Examination [19] | M-ACE |

[a]Adapted version of the Halstead-Reitan Trail Making Test [16]

[b]Adapted version of the NIH Toolbox Picture Vocabulary Test [13]

[c]Adapted version of the Test the Nation Paired Associate Learning Test [30]

[d]Adapted version of the one-touch Tower of London Test [31]

[e]Adapted version of the COGNITO Matrices test [23]

**Demographic and health questionnaire.** Information on age, sex, and education was collected. Participant's age was calculated from their date of birth. To measure education, participants were asked, "How many years of full-time education have you completed?". General health was assessed by asking participants, "In general, would you say your health is excellent, very good, good, fair, or poor", and "Compared to one year ago, how you would rate your health in general now?". Participants selected from the following answers: Much better now than one year ago; somewhat better now than one year ago; about the same; somewhat worse now than one year ago; much worse now than one year ago.

**UK Biobank cognitive assessment questionnaire.** The UK Biobank cognitive test battery was designed to be administered unsupervised, and, although there were staff members present in the UK Biobank clinic during test administration, participants were expected to work through the cognitive assessment independently without a tester observing. Because no tester was there to help participants understand the test instructions, the onscreen instructions for each test must be clear. To assess this, after completing the UK Biobank cognitive assessment

participants were asked, "Did you generally find that the instructions for all the tasks were clear?". Participants answered either yes or no. Next, participants were shown screenshots of each UK Biobank cognitive test and were asked "Were the instructions for this test clear?". Participants answered either yes or no.

The UKB Numeric Memory task was designed to assess backward digit span, which involves participants remembering a sequence of digits and then mentally reversing them in their mind. However, for this task, all the numbers in the to-be-remembered sequence were presented on the screen at once. This meant that some individuals were able to get the correct answer by reading the number sequence from right-to-left and not reversing the digits in their mind. These individuals are actually performing a forward digit span, which is an easier task. To identify the number of participants who completed this task forwards or backwards, participants were asked how they completed this task. The possible options were: read from left-to-right and reversed the digits in your mind; read from right-to-left and did not need to reverse digits in your mind; a mixture of both; something different.

## Procedure

Study visits took place in the Psychology Department at the University of Edinburgh at a time mutually agreed by the participant and the tester. Appointments were available in the morning, afternoon, or evening on both weekdays and weekends, to suit the participant's schedule. All assessments were administered by the same psychology-graduate tester (CF-R) in a quiet room, one-to-one, free of distractions. After reading the information sheet and signing the consent form, participants were administered the demographic and health questionnaire, and then the self-rated memory questionnaire. The testing session took approximately 2.5 to 3 hours to complete. To limit any effects of fatigue on test performance, the test order was counter-balanced. Individuals with even participant ID numbers completed the UK Biobank tests before completing the M-ACE and reference tests. Individuals with odd participant ID numbers completed the M-ACE and reference tests first and then completed the UK Biobank tests. The UK Biobank questionnaire was administered immediately after completing the UK Biobank cognitive assessment. Approximately half-way through the session, participants were given a short, approximately 15–20 minute break (with refreshments), again to try to limit any effects of fatigue. The test order for participants with odd and even ID numbers is shown in S1 File (S2 Table).

We aimed to recruit a subsample of 50 participants to come back approximately four weeks after the first visit and repeat the UK Biobank cognitive assessment for a second time. Participants who indicated on the consent form that they would be willing to return for a second study visit, and who were able to arrange an appointment four weeks (± 1 week) after the first assessment, were invited back to complete the UK Biobank cognitive assessment again. Individuals who agreed to return for a second visit were administered the UK Biobank questionnaire after completing the UK Biobank tests for a second time.

## Statistical analyses

All analyses were performed in R version 3.5.2. To examine the association between UK Biobank cognitive tests, general tests, and reference tests with basic demographic characteristics, correlations were calculated between all cognitive tests and age, sex, years of education, and self-reported general health. For all correlational analyses reported in this paper, both Pearson *r* and Spearman *rho* correlations were calculated. Point-biserial correlations were calculated for correlations with sex and UKB Prospective Memory, as these are binary variables. Concurrent validities of the UK Biobank tests were calculated by correlating the UK Biobank cognitive

tests with the general tests and reference tests. Partial correlations, adjusting for age, were also calculated to determine whether the sizes of the associations between the UK Biobank cognitive tests and the general and reference tests remained after controlling for age. The correlation between scores on WMS-IV Designs I and II was high ($r = 0.65$, $p < .001$); therefore, a total score was created by summing the scores on Designs I and Designs II (WMS-IV Designs Total; max score = 240). Similarly, the correlation between WMS-IV VPA I and II was high ($r = 0.89$, $p < .001$); therefore, a total VPA score was created by summing the scores on VPA I and VPA II (WMS-IV VPA Total; maximum score = 70). Methods to adjust for multiple testing were not used here. This study was interested in the size of the associations between different cognitive tests, and was less interested in whether these associations were statistically significant.

To investigate whether a measure of general cognitive ability created using UK Biobank cognitive tests was highly correlated with a measure of general cognitive ability created using some of the well-validated reference tests, three measures of general cognitive ability were created using the following combinations of tests: 1) all UK Biobank tests included in the enhanced assessment administered at the imaging study; 2) the UK Biobank baseline tests; and 3) a selection of the reference tests. Some of the cognitive tests have multiple parts that are highly correlated (e.g., TMT part A and TMT part B; DLRT Simple RT and DLRT Choice RT). Only one score from each cognitive test was used to create each measure of general cognitive ability to ensure that highly correlated parts of tests do not overly influence the general cognitive ability score. For DLRT, the Choice RT part was chosen as this is a more cognitively challenging task than Simple RT. For TMT and UKB TMT, part B was used because TMT is thought to be a test of executive function—specifically switching ability—and part B is thought to assess this switching ability, whereas part A is often thought to be assessing processing speed.

Each of the measures of general cognitive ability were created by entering cognitive tests scores into a PCA, checking the eigenvalues and scree plots, and saving the scores on the first unrotated principal component. Before cognitive test scores were entered into the PCA, test distributions (S1 File, S1 Fig) were inspected and, where possible, scores with non-normal distributions were transformed. The specific transformations performed are described below. To reduce the influence of any outliers, tests scores were winsorized to 3 SD.

## General cognitive ability—Using 11 reference tests

Scores on the following reference tests were entered into a PCA: TMT part B (log-transformed), SDMT, WMS-IV Designs Total, WAIS-IV Digit Span Total (created by summing scores on WAIS-IV Digit Span Forward and Digit Span Backward), D-KEFS Tower Test, DLRT Choice RT (log-transformed), NIH Toolbox Picture Vocabulary, NART (scores were reverted and log-transformed), PPVT (scores were reverted and log-transformed), WMS-IV VPA Total, and COGNITO Matrices score. This measure of general cognitive ability was designed to reflect a *g* component created using well-validated and comprehensive cognitive tests that have been viewed as the 'gold standard' cognitive measures. As the RMBM Appointments test is brief and contains only 2 items, this test was not included in this comprehensive measure of general cognitive ability. Eigenvalues and scree plot (S1 File, S2 Fig) indicated two components. These two components accounted for 56% of the variance in the 11 reference cognitive tests. Test loadings (unrotated and rotated using oblique rotation) are shown in S1 File (S3 Table). Inspection of the rotated loadings suggests that the first component appears to reflect processing speed. The tests which load most highly on this component include TMT part B (-0.83), SDMT (0.82), WAIS-IV Designs Total (0.73), and DLRT Choice RT (-0.71). Non-speeded verbal tests load highly on the second component. The loadings for the NART,

NIH Toolbox Picture Vocabulary, and PPVT were 0.91, 0.90, and 0.84, respectively. Scores on the first unrotated principal component, which accounted for 35% of the total variance, were saved and used as a measure of general cognitive ability (*g:reference-11*).

## General cognitive ability—Using 11 UK Biobank cognitive tests

Scores on the following tests were entered into a PCA: UKB Pairs Memory (log (x+1) transformed), UKB RT (log-transformed), UKB Prospective Memory, UKB Fluid IQ, UKB Numeric Memory, UKB TMT part B (log transformed), UKB Symbol Digit, UKB Picture Vocabulary, UKB Paired Associate Learning, UKB Tower Test, and UKB Matrices. Eigenvalues and scree plot (S1 File, S3 Fig) indicated two components. These two components accounted for 46% of the variance in the 11 UK Biobank cognitive tests. Test loadings (unrotated and rotated using oblique rotation) are shown in S1 File (S4 Table). Like the results from the PCA for the reference tests, the first rotated component from the UKB tests appears to reflect processing speed, whereas the second component reflects non-speeded and verbal abilities. Examining the rotated loadings, tests that load highly on the first component include UKB TMT part B (-0.82), UKB Symbol Digit (0.80), and UKB Tower Test (0.71). UKB Picture Vocabulary loads highly on the second component (0.91). UKB PAL (0.48) and UKB Fluid IQ (0.45)—two verbal tests—also load moderately highly on the second component. Scores on the first unrotated principal component, which accounted for 34% of the total variance, were saved and used as a measure of general cognitive ability (*g:UKB-11*).

The results of the PCA of 11 reference tests, and the PCA of 11 UK Biobank tests were generally similar, with the first rotated component reflecting processing speed and the second reflecting non-speeded, verbal ability. The first unrotated principal component (i.e., *g*) also accounted for a similar proportion of the variance in the test scores (35% and 34% for the reference and the UK Biobank tests, respectively). As a result of the fact that many of the tests used to create these *g* components were speeded tests, these measures of general cognitive ability we have created are largely measuring speeded/fluid cognitive abilities. We also note that only one vocabulary test was used to create *g:UKB-11*, whereas three were used in the creation of *g:reference-11*. This discrepancy is likely to be an important reason why the loading on the first unrotated principal component for UKB Picture Vocabulary (0.19) is lower than for NIH Toolbox Picture Vocabulary (0.51).

**General cognitive ability—Using 5 UK Biobank cognitive tests.** Scores on the following tests were entered into a PCA: UKB Pairs Memory (log (x+1) transformed), UKB RT (log-transformed), UKB Prospective Memory, UKB Fluid IQ, and UKB Numeric Memory. Eigenvalues and scree plot (S1 File, S4 Fig) indicated one component. This component accounted for 38% of the total variance in the 5 tests. The test loadings are reported in S1 File (S5 Table). Scores on this unrotated principal component were saved and used as a measure of general cognitive ability *(g:UKB-5)*.

Correlations between the three measures of general cognitive ability were calculated. The correlations and age-adjusted correlations between *g:reference-11* and each of the UK Biobank cognitive tests were calculated. We also calculated the correlations and age-adjusted correlations between *g:UKB-11* and *g:UKB-5* with each of the general and reference tests.

To investigate whether participants thought the instructions for the UK Biobank cognitive tests were clear, the number and percentage of participants who answered 'no' to "Did you generally find that the instructions for all the tasks were clear?" was calculated. Next, the number and percentage of participants who reported 'no' when asked whether the instructions for each individual UK Biobank test were clear was calculated.

The number and percentage of participants who reported carrying out a forward digit span, a backward digit span, a mixture of both, or something else when completing UKB Numeric Memory was calculated. Between-group analysis of variance (ANOVA) was used to determine whether mean performance on the UKB Numeric Memory test differed by technique reported.

To measure the short-term stability of the UK Biobank tests, Pearson and Spearman test-retest correlations were calculated between scores on the UK Biobank tests at Time 1 and Time 2.

## Results

Participant characteristics are reported in S1 File (S6 Table). A total of 160 participants (mean age = 62.59, SD = 10.24) completed the full assessment at Time 1. Of these, 52 participants (mean age = 61.69, SD = 9.70) returned and repeated the UK Biobank tests at Time 2. The mean time to repeat was 28.88 days (SD = 2.02, range = 26 to 36). The sample used here were relatively highly educated (mean years of full-time education = 16.19, SD = 2.73), and most reported their health to be very good (n = 85; 53.1%) or excellent (n = 36, 22.5%). Extended descriptive statistics (n, mean, SD and range) for each of the cognitive tests administered in this study are reported in S1 File (S7 Table).

For all correlations reported throughout this report, both Pearson correlations and Spearman rank-order correlations were calculated. These correlations tended to be very similar, therefore only the Pearson correlations are reported in the main text. Spearman correlations are reported in the supplementary materials. All correlations carried out for this study, and their exact p-values, are reported in S1 Table (Pearson correlations; S8 Table) and S2 Table (Spearman rank-order correlations; S9 Table).

### Correlations between cognitive test scores and demographic and health variables

The Pearson correlations between each of the UK Biobank tests and age is shown in Table 2. Note that, UKB RT, UKB TMT part A, and UKB TMT part B are measuring response times. On these tests, higher scores indicate that the participants took longer to complete the tests and therefore higher scores reflect poorer performance. The score for UKB Pairs Matching is the number of errors made matching all of the cards and, therefore, a higher score also reflects poorer performance. For all other UK Biobank tests, higher scores indicate better performance. All UK Biobank tests correlated significantly with age. In all but one, older individuals performed more poorly on these tests (absolute $r$ = 0.16 to 0.60, $p \leq .040$). The exception was UKB Picture Vocabulary where older participants performed better than younger participants on this test ($r$ = 0.18, $p$ = .022). The strongest age associations were seen for tests measuring processing speed. Older adults tended to have lower scores on UKB Symbol Digit ($r$ = -0.60, $p < .001$), and were slower on UKB TMT part A ($r$ = 0.58, $p < .001$), and UKB TMT part B ($r$ = 0.57, $p < .001$). UKB Pairs Matching ($r$ = 0.34), UKB Tower Test ($r$ = -0.45), and UKB Matrices ($r$ = -0.47) also had absolute correlations of > 0.3 (for all, $p < .001$) with age.

The Pearson and Spearman rank-order correlations between all cognitive tests and age, sex, years of education, and general health are shown in S1 File (S10 and S11 Tables). Male participants had lower scores than female participants on UKB PAL ($r$ = -0.28, $p < .001$), but higher scores on the UKB Tower Test ($r$ = 0.19, $p$ = .018) and UKB Matrices ($r$ = 0.17, $p$ = .034). Individuals with more years of education were quicker on UKB TMT parts A ($r$ = -0.18, $p$ = .024) and B ($r$ = -0.21, $p$ = .007), and scored higher on UKB Picture Vocabulary ($r$ = 0.29, $p < .001$) and UKB Matrices ($r$ = 0.32, $p < .001$). None of the UK Biobank tests were associated with general health.

**Table 2. Pearson correlations between UK Biobank tests and age, general tests, and reference tests (n = 154–160).**

| | UKB Pairs Matching | UKB RT | UKB Prosp Memory | UKB Fluid IQ | UKB Numeric Memory | UKB TMT part A | UKB TMT part B | UKB Symbol Digit | UKB Picture Vocabulary | UKB PAL | UKB Tower Test | UKB Matrices |
|---|---|---|---|---|---|---|---|---|---|---|---|---|
| Age | 0.339*** | 0.253** | -0.268** | -0.229** | -0.207** | 0.575*** | 0.565*** | -0.598*** | 0.181* | -0.163* | -0.454*** | -0.474*** |
| *General tests* | | | | | | | | | | | | |
| Self-rated memory | 0.181* | 0.144 | -0.176* | -0.169* | -0.157 | 0.307*** | 0.110 | -0.239** | -0.043 | -0.106 | -0.097 | -0.061 |
| M-ACE | -0.305*** | -0.041 | 0.270** | 0.351*** | 0.338*** | -0.252** | -0.285*** | 0.254** | 0.376*** | 0.468*** | 0.200* | 0.218** |
| *Reference tests* | | | | | | | | | | | | |
| PPVT | -0.113 | -0.147 | 0.267** | 0.362*** | 0.169* | -0.045 | -0.124 | 0.008 | **0.744***** | 0.256** | 0.094 | 0.264** |
| NART | -0.010 | 0.015 | 0.102 | 0.291*** | 0.148 | 0.007 | -0.028 | -0.133 | **0.748***** | 0.316*** | -0.053 | 0.119 |
| NIH TB Picture Vocabulary | -0.101 | -0.044 | 0.231** | 0.346*** | 0.135 | -0.037 | -0.021 | -0.057 | **0.826***** | 0.326*** | 0.061 | 0.183* |
| RMBM Appointments | -0.073 | -0.136 | **0.224**** | 0.179* | 0.123 | -0.085 | -0.176* | 0.100 | 0.119 | 0.229** | 0.148 | 0.087 |
| WMS-IV Designs Total | **-0.331***** | -0.243** | 0.341*** | 0.303*** | 0.307*** | -0.387*** | -0.497*** | 0.484*** | 0.006 | 0.342*** | 0.448*** | 0.432*** |
| WMS-IV VPA Total | -0.296*** | -0.033 | 0.363*** | 0.211** | 0.328*** | -0.210** | -0.341*** | 0.210** | 0.316*** | **0.470***** | 0.185* | 0.275*** |
| WAIS-IV Digit Span Forwards | -0.066 | -0.067 | 0.234** | 0.369*** | **0.434***** | -0.213** | -0.220** | 0.165* | 0.146 | 0.047 | 0.267** | 0.187* |
| WAIS-IV Digit Span Backwards | -0.118 | -0.048 | 0.265** | 0.430*** | **0.510***** | -0.223** | -0.221** | 0.177* | 0.287*** | 0.196* | 0.234** | 0.222** |
| WAIS-IV Digit Span Sequence | -0.122 | -0.080 | 0.297*** | 0.460*** | **0.417***** | -0.268** | -0.392*** | 0.300*** | 0.204* | 0.298*** | 0.381*** | 0.304*** |
| SDMT | -0.268** | -0.353*** | 0.255** | 0.369*** | 0.357*** | -0.465*** | -0.540*** | **0.636***** | -0.001 | 0.243** | 0.493*** | 0.424*** |
| DLRT Simple RT | 0.090 | **0.521***** | -0.189* | -0.253** | -0.324*** | 0.218** | 0.213** | -0.249** | 0.014 | -0.161* | -0.196* | -0.230** |
| DLRT Choice RT | 0.122 | **0.431***** | -0.334*** | -0.289*** | -0.283*** | 0.425*** | 0.514*** | -0.470*** | -0.041 | -0.203* | -0.339*** | -0.360*** |
| TMT part A | 0.214** | 0.225** | -0.226** | -0.155 | -0.266** | **0.439***** | 0.499*** | -0.519*** | 0.218** | -0.121 | -0.387*** | -0.232** |
| TMT part B | 0.277*** | 0.248** | -0.306*** | -0.303*** | -0.481*** | 0.461*** | **0.663***** | -0.558*** | -0.042 | -0.345*** | -0.451*** | -0.383*** |
| D-KEFS Tower Test | -0.398*** | -0.116 | 0.302*** | 0.275*** | 0.241** | -0.280*** | -0.372*** | 0.409*** | 0.052 | 0.245** | **0.402***** | 0.360*** |
| COGNITO Matrices | -0.380*** | -0.149 | 0.408*** | 0.383*** | 0.380*** | -0.354*** | -0.364*** | 0.414*** | 0.248** | 0.256** | 0.371*** | **0.574***** |

Correlations shown in bold show the correlations between the UK Biobank tests and the chosen reference test.

UKB, UK Biobank; RT, Reaction Time; Prosp Memory, Prospective Memory; Fluid IQ, Fluid Intelligence; TMT, Trail Making Test; PAL, Paired Associate Learning; M-ACE, Mini Addenbrooke's Cognitive Examination; PPVT, Peabody Picture Vocabulary Test–Fourth Edition; NART, National Adult Reading Test; NIH TB Picture Vocabulary, NIH Toolbox Picture Vocabulary; RMBM, Rivermead Behavioural Memory Test–Extended Version; WMS-IV, Wechsler Memory Scale–Fourth Edition; VPA, Verbal Paired Associates; WAIS-IV, Wechsler Adults Intelligence Test–Fourth Edition; SDMT, Symbol Digit Modalities Test; DLRT, Deary-Liewald Reaction Time Test; D-KEFS, Delis-Kaplan Executive Function System.

*$p < .05$,

**$p < .01$,

***$p < .001$

## Associations with general tests

The Pearson correlations between the UK Biobank tests and the general tests are reported in Table 2. The Spearman rank-order correlations are reported in S1 File (S12 Table). Reporting poorer self-rated memory was associated with more errors on UKB Pairs Matching ($r = 0.18$, *p*

= .022), being less likely to correctly touch the orange circle in UKB Prospective Memory ($r$ = -0.18, $p$ = .026), having lower scores on UKB Fluid IQ ($r$ = -0.17, $p$ = .033) and UKB Symbol Digit ($r$ = -0.24, $p$ = .002), and being slower on UKB TMT part A ($r$ = 0.31, $p$ < .001).

Except for UKB RT, higher scores on the M-ACE were associated with better performance on all UK Biobank cognitive tests, with absolute effect sizes of 0.2 or higher. The M-ACE was most strongly correlated with performance on UKB PAL ($r$ = 0.47, $p$ < .001). UKB Picture Vocabulary ($r$ = 0.38), UKB Fluid IQ ($r$ = 0.35), UKB Numeric Memory ($r$ = 0.34), and UKB Pairs Matching ($r$ = -0.31) also correlated moderately with the M-ACE (for all, $p$ < .001).

## Associations with reference tests

The Pearson correlations between the UK Biobank cognitive tests and the reference tests are reported in Table 2. The Spearman rank-order correlations are reported in S1 File (S12 Table). The correlations highlighted in bold in Table 2 (and S1 File, S12 Table) reflect the correlations between the UK Biobank test and the chosen reference test(s) which was judged to be assessing the same cognitive capability or domain. In going through each UK Biobank cognitive test's results below we first describe the correlation with the respective reference test(s), and then we highlight correlations with 'non-reference' tests that have absolute effect sizes greater than 0.3.

**UKB Pairs Matching.** Better performance on UKB Pairs Matching was associated with higher scores on WMS-IV Designs Total—the reference test for UKB Pairs Matching ($r$ = -0.33, $p$ < .001). Better performance on the UKB Pairs Matching test was also moderately associated with better performance on D-KEFS Tower Test ($r$ = -0.40, $p$ < .001) and COG-NITO Matrices ($r$ = -0.38, $p$ < .001).

**UKB RT.** The DLRT Simple and Choice RT were chosen as reference tests for UKB RT. Slower response on UKB RT was associated with slower responses on DLRT Simple RT ($r$ = 0.52, $p$ < .001) and DLRT Choice RT ($r$ = 0.43, $p$ < .001). The SDMT, another measure of processing speed, also correlated moderately with UKB RT such that individuals who had quicker responses on UKB RT scored higher, and were therefore quicker, on the SDMT ($r$ = -0.35, $p$ < .001).

**UKB Prospective Memory.** There was a small, positive correlation between UKB Prospective Memory and the chosen reference test, RMBM Appointments ($r$ = 0.22, $p$ = .005). All other reference tests, except NART and DLRT Simple RT, had stronger correlations with UKB Prospective Memory than that reported between UKB Prospective Memory and RMBM Appointments. Correctly answering the UKB Prospective Memory test on the first attempt was most strongly associated with higher scores on COGNTIO Matrices ($r$ = 0.41), and had moderate correlations with better performance on WMS-IV VPA Total ($r$ = 0.36), WMS-IV Designs total ($r$ = 0.34), DLRT Choice RT ($r$ = -0.33), TMT part B ($r$ = -0.31), and D-KEFS Tower Test ($r$ = 0.30) (for all, $p$ < .001).

**UKB Fluid IQ.** Higher UKB Fluid IQ scores correlated most strongly with higher WAIS-IV Digit Span scores (Forwards $r$ = 0.37; Backwards $r$ = 0.43; Sequence $r$ = 0.46) (for all, $p$ < .001). Higher UKB Fluid IQ score also correlated moderately with better performance on COG-NITO Matrices ($r$ = 0.38), SDMT ($r$ = 0.37), PPVT ($r$ = 0.36), NIH Toolbox Picture Vocabulary ($r$ = 0.35), WMS-IV Designs Total ($r$ = 0.30), and TMT part B ($r$ = -0.30) (for all, $p$ < .001).

**UKB Numeric Memory.** This test correlated positively with the chosen reference test—WAIS-IV Digit Span. The correlations for WAIS-IV Digit Span Forwards, Backwards and Sequence were 0.43, 0.51, and 0.42, respectively (for all, $p$ < .001). UKB Numeric Memory also correlated moderately with TMT part B, such that individuals who scored higher on UKB Numeric Memory were quicker on TMT part B ($r$ = -0.48, $p$ < .001). Higher scores on UKB Numeric Memory had associations > 0.3 with better performance on COGNITO Matrices

($r$ = 0.38), SDMT ($r$ = 0.36), WMS-IV VPA Total ($r$ = 0.33), DLRT Simple RT ($r$ = -0.32), and WMS-IV Designs Total ($r$ = 0.31) (for all, $p$ < .001).

**UKB TMT part A.** UKB TMT part A correlated positively with TMT part A ($r$ = 0.44, $p$ < .001)—the chosen reference test—and TMT part B ($r$ = 0.46, $p$ < .001). Faster completion of UKB TMT part A also moderately correlated with better performance on the SDMT ($r$ = -0.47), DLRT Choice RT ($r$ = 0.43), WMS-IV Designs Total ($r$ = -0.39), and COGNITO Matrices ($r$ = -0.35) (for all, $p$ < .001).

**UKB TMT part B.** UKB TMT part B was strongly and positively correlated with the paper-and-pencil version of the TMT part B (the reference test; $r$ = 0.66, $p$ < .001), and also with TMT part A ($r$ = 0.50, $p$ < .001). Being quicker on UKB TMT part B was also moderately correlated with better performance on SDMT ($r$ = -0.54), DLRT Choice RT ($r$ = 0.51), WMS-IV Designs Total ($r$ = -0.50), WAIS-IV Digit Span Sequence ($r$ = -0.39), D-KEFS Tower Test ($r$ = -0.37), COGNITO Matrices ($r$ = -0.36), and WMS-IV VPA Total ($r$ = -0.34) (for all, $p$ < .001).

**UKB Symbol Digit.** Higher scores on UKB Symbol Digit were correlated at $r$ = 0.64 ($p$ < .001) with higher scores on the SDMT, the reference test for UKB Symbol Digit. Better performance on this test also correlated at > 0.3 with better performance on TMT part A ($r$ = -0.52) and part B ($r$ = -0.56), WMS-IV Designs Total ($r$ = 0.48), DLRT Choice RT ($r$ = -0.47), COGNITO Matrices ($r$ = 0.41), and D-KEFS Tower Test ($r$ = 0.41) (for all, $p$ < .001).

**UKB Picture Vocabulary.** UKB Picture Vocabulary correlated highly with its three reference tests: 0.83 with the NIH Toolbox Picture Vocabulary, 0.75 with the NART, and 0.74 with PPVT (for all, $p$ < .001). Higher scores on UKB Picture Vocabulary were associated with higher scores on WMS-IV VPA Total score ($r$ = 0.32, $p$ < .001); a test with verbal contents.

**UKB PAL.** UKB PAL was positively correlated with the WMS-IV VPA Total, the reference test for UKB PAL ($r$ = 0.47, $p$ < .001). Higher UKB PAL scores also correlated at > 0.3 with better performance on TMT part B ($r$ = -0.35), WMS-IV Designs Total ($r$ = 0.34), NIH Picture Vocabulary ($r$ = 0.33), and NART ($r$ = 0.32) (for all, $p$ < .001).

**UKB Tower Test.** UKB Tower Test correlated positively with its reference test, the D-KEFS Tower Test ($r$ = 0.40, $p$ < .001). Better performance on the UKB Tower Test was also associated with better performance on SDMT ($r$ = 0.49), TMT part B ($r$ = -0.45), WMS-IV Designs Total ($r$ = 0.45), TMT part A ($r$ = -0.39), WAIS-IV Digit Span Sequence ($r$ = 0.38), COGNITO Matrices ($r$ = 0.37), and DLRT Choice RT ($r$ = -0.34) (for all, $p$ < .001).

**UKB Matrices.** This test correlated positively at $r$ = 0.57 ($p$ < .001) with the original version of this test; the COGNITO Matrices. Higher performance on this test also correlated moderately with better performance on WMS-IV Designs Total ($r$ = 0.43), SDMT ($r$ = 0.42), TMT part B ($r$ = -0.38), D-KEFS Tower Test ($r$ = 0.36), DLRT Choice RT ($r$ = -0.36), and WAIS-IV Digit Span Sequence ($r$ = 0.30) (for all, $p$ < .001).

**Partial correlations adjusting for age.** The age-adjusted Pearson and Spearman correlations between the UK Biobank tests and the general and reference tests are reported in S1 File (S13 Table for the age-adjusted Pearson correlations; S14 Table for the age-adjusted Spearman correlations). After controlling for age, none of the correlations between self-rated memory and the UK Biobank tests were significant. The exception was the correlation between self-rated memory and UKB TMT part A, which remained significant, though reduced in size ($r$ = 0.31; age-adjusted $r$ = 0.17). All correlations between M-ACE and the UK Biobank tests were smaller—though most remained significant—when adjusting for age, except for the correlation between M-ACE and UKB Picture Vocabulary, which became stronger ($r$ = 0.38; age-adjusted $r$ = 0.43). The correlations between UKB Tower Test ($r$ = 0.20; age-adjusted $r$ = 0.12) and UKB Matrices ($r$ = 0.22; age-adjusted $r$ = 0.14) with the M-ACE were no longer significant when adjusting for age.

Generally, the age-adjusted correlations between the UK Biobank tests and the reference tests tended to be smaller than the raw correlations, though the difference between the raw correlations and the age-adjusted correlations was small. The largest differences were seen for the correlations between the following UK Biobank tests with their respective reference tests: UKB TMT part B ($r = 0.66$; age-adjusted $r = 0.55$), UKB Symbol Digit ($r = 0.64$; age-adjusted $r = 0.45$), UKB Pairs Matching ($r = -0.33$; age-adjusted $r = -0.19$), and UKB TMT part A ($r = 0.44$; age-adjusted $r = 0.24$). For all other correlations between UK Biobank cognitive tests and the reference tests, the change in the strength of the correlation between the raw correlations and the age-adjusted correlations was $\leq 0.07$.

## Measures of general cognitive ability

The Pearson correlation between a measure of general cognitive ability created using 11 well-validated reference tests (*g:reference-11*) and 11 UK Biobank tests (*g:UKB-11*) was $r = 0.83$ ($p < .001$; age-adjusted $r = 0.79$, $p < .001$). The correlation was similar when re-run using a measure of general cognitive ability that was created excluding scores on the COGNITO Matrices and NIH Toolbox Picture Vocabulary test, which share items with UKB Matrices and UKB Picture Vocabulary (see S1 File). The correlation between *g:reference-11* and a measure of general cognitive ability created using the five UK Biobank baseline tests (*g:UKB-5*) was $r = 0.74$ ($p < .001$; age-adjusted $r = 0.69$, $p < .001$).

**Correlations between *g:reference-11* and UK Biobank tests.** Pearson correlations and age-adjusted Pearson correlations between *g:reference-11* and each of the UK Biobank tests are reported in Table 3 (Spearman rank-order correlations are reported in S1 File, S16 Table). All UK Biobank cognitive tests correlated with this measure of general cognitive ability, such that higher scores on *g:reference-11* were associated with better performance on the UK Biobank cognitive tests (for all, $p < .001$). UKB RT had the lowest correlation with *g:reference-11* ($r = -0.29$, $p < .001$), whereas UKB TMT part B had the strongest correlation ($r = -0.62$, $p < .001$).

**Table 3. Pearson correlations and age-adjusted Pearson correlations between general cognitive ability created using 11 reference tests and the UK Biobank tests (n = 151–160).**

| | *g:reference-11* | |
| --- | --- | --- |
| | *r* | Age-adjusted *r* |
| UKB Pairs Matching Test | -0.402*** | -0.293*** |
| UKB Reaction Time Test | -0.294*** | -0.214** |
| UKB Prospective Memory Test | 0.487*** | 0.420*** |
| UKB Fluid Intelligence Test | 0.553*** | 0.519*** |
| UKB Numeric Memory Test | 0.551*** | 0.533*** |
| UKB Trail Making Test part A | -0.494*** | -0.315*** |
| UKB Trail Making Test part B | -0.621*** | -0.498*** |
| UKB Symbol Digit Test | 0.536*** | 0.366*** |
| UKB Picture Vocabulary Test | 0.430*** | 0.578*** |
| UKB Paired Associate Learning Test | 0.479*** | 0.460*** |
| UKB Tower Test | 0.521*** | 0.394*** |
| UKB Matrix Pattern Test | 0.581*** | 0.462*** |

*g:reference-11*, measure of general cognitive ability created by entering 11 standardised reference tests into a principal component analysis and saving scores from the first unrotated principal component; UKB, UK Biobank.

*$p < .05$,

**$p < .01$,

***$p < .001$

Other UK Biobank tests which correlated positively with general cognitive ability at $> 0.5$ were UKB Matrices ($r = 0.58$), UKB Fluid IQ ($r = 0.55$), UKB Numeric memory ($r = 0.55$), UKB Symbol Digit ($r = 0.54$), and UKB Tower Test ($r = 0.52$) (for all, $p < .001$).

The age-adjusted Pearson correlations between *g:reference-11* and UK Biobank tests (Table 3) tended to be weaker than the raw correlations, except for the correlation between *g: reference-11* and UKB Picture Vocabulary, which became stronger (raw $r = 0.43$, $p < .001$; age-adjusted $r = 0.58$, $p < .001$).

**Correlations between *g:UKB-11* and *g:UKB-5* with the general and reference tests.**
Pearson correlations and age-adjusted Pearson correlations between general cognitive ability created using the UK Biobank tests (*g:UKB-11* and *g:UKB-5*) and the general tests and reference tests are shown in Table 4 (Spearman rank-order correlations are reported in S1 File, S17 Table). Higher scores on *g:UKB-11* were associated with better performance on all the general and reference tests, except for self-rated memory and NART which were not significantly associated with *g:UKB-11*. Higher *g:UKB-11* score was most strongly related to better performance on SDMT ($r = 0.68$), TMT part B ($r = -0.64$), and WMS-IV Designs Total ($r = 0.63$) (for all, $p < .001$).

When adjusting for age, some of the associations between *g:UKB-11* and the reference tests reduced in strength (e.g., correlations with tests of speed, executive function, and reasoning),

**Table 4. Pearson correlations and age-adjusted Pearson correlations between two measures of general cognitive ability, created using the UK Biobank cognitive tests, and the general and reference tests (n = 151–160).**

|  | *g:UKB-11* | | *g:UKB-5* | |
|---|---|---|---|---|
|  | *r* | Age-adjusted *r* | *r* | Age-adjusted *r* |
| Self-rated memory | -0.259 | -0.106 | -0.283*** | -0.181* |
| M-ACE | 0.427*** | 0.416*** | 0.418*** | 0.380*** |
| PPVT | 0.263** | 0.423*** | 0.326*** | 0.407*** |
| National Adult Reading Test | 0.097 | 0.354*** | 0.151 | 0.305*** |
| NIH Toolbox Picture Vocabulary | 0.188* | 0.384*** | 0.251** | 0.359*** |
| RMBM Appointments | 0.244** | 0.264** | 0.250** | 0.237** |
| WMS-IV Designs Total | 0.629*** | 0.481*** | 0.496*** | 0.384*** |
| WMS-IV VPA Total | 0.427*** | 0.357*** | 0.395*** | 0.334*** |
| WAIS-IV Digit Span Forwards | 0.359*** | 0.366*** | 0.405*** | 0.394*** |
| WAIS-IV Digit Span Backwards | 0.371*** | 0.397*** | 0.456*** | 0.453*** |
| WAIS-IV Digit Span Sequence | 0.488*** | 0.522*** | 0.445*** | 0.436*** |
| Symbol Digit Modalities Test | 0.681*** | 0.522*** | 0.511*** | 0.387*** |
| DLRT Simple Reaction Time | -0.376*** | -0.353*** | -0.438*** | -0.408*** |
| DLRT Choice Reaction Time | -0.567*** | -0.432*** | -0.468*** | -0.362*** |
| Trail Making Test part A | -0.493*** | -0.313*** | -0.351*** | -0.213** |
| Trail Making Test part B | -0.639*** | -0.514*** | -0.502*** | -0.395*** |
| D-KEFS Tower Test | 0.486*** | 0.437*** | 0.416*** | 0.355*** |
| COGNITO Matrices | 0.599*** | 0.539*** | 0.539*** | 0.479*** |

*g:UKB-11*, measure of general cognitive ability created by entering 11 UK Biobank cognitive tests into a principal component analysis and saving scores from the first unrotated principal component; *g:UKB-5*, measure of general cognitive ability created by entering the 5 UK Biobank tests administered at the UK Biobank baseline assessment into a principal component analysis and saving scores from the first unrotated principal component; M-ACE, Mini Addenbrooke's Cognitive Examination; PPVT, Peabody Picture Vocabulary Test–Fourth Edition; RMBM, Rivermead Behavioural Memory Test–Extended Version; WMS-IV, Wechsler Memory Scale–Fourth Edition; WAIS-IV, Wechsler Adult Intelligence Scale, Fourth Edition; DLRT, Deary-Liewald Reaction Time Test; D-KEFS, Delis-Kaplan Executive Function System.

*$p < .05$,

**$p < .01$,

***$p < .001$

whereas others became stronger (e.g., correlations with vocabulary tests, RMBM Appointments, and WAIS-IV Digit Span). However, when adjusting for age, all tests except self-rated memory were associated with *g:UKB-11* such that a better *g:UKB-11* score was associated with better test scores on the general and reference tests. Whereas there was no association between *g:UKB-11* and the NART when calculating raw correlations ($r = 0.10$, $p > .05$), there was a moderate and positive association between *g:UKB-11* and the NART when adjusting for age (age-adjusted $r = 0.35$, $p < .001$).

Higher scores on *g:UKB-5* were also associated with better performance on the general and reference tests, except the NART, which was not associated with *g:UKB-5*. Again, the association between *g:UKB-5* and the NART became significant when adjusting for age ($r = 0.31$, $p < .001$). Generally, the correlations seen between *g:UKB-5* and the reference tests were lower than those seen between *g:UKB-11* and the reference tests.

## UK Biobank questionnaire

**Clear test instructions.** The number and percentage of participants who thought the UK Biobank tests were unclear is reported in S1 File (S18 Table). A total of 8 (5.5%) participants reported that they thought the UK Biobank test instructions in general were unclear. Nearly one quarter of participants (n = 35, 24.1%) reported that they thought the instructions for the UKB Tower Test were not clear. Participants generally thought the instructions for UKB RT, UKB Picture Vocabulary and UKB Matrices were clear. Only 3 (2.1%), 2 (1.4%), and 1 (0.7%) participants, respectively, reported that the instructions for these tests were not clear.

**UKB numeric memory technique.** Of the 141 individuals who were asked about the technique used to complete UKB Numeric Memory, only 20 (14.2%) participants reported that they performed the UKB Numeric Memory test as a backward digit span (e.g., read from left-to-right and reversed the digits in their mind). Most (n = 102; 72.3%) performed a forward digit span (e.g., read from right-to-left and did not reverse the digits in their mind). The remaining participants (n = 19; 13.5%) reported using a mixture of both techniques. Participants who did a backward digit span (mean = 6.70, SD = 1.08) had a slightly lower mean score on UKB Numeric Memory than those who did a forward digit span (mean = 6.92, SD = 1.24) or those who did a mixture of both (mean = 7.11, SD = 1.25). A between-group ANOVA did not reveal any differences in UKB Numeric Memory scores by technique used (F (2, 136) = 0.518, $p = .602$).

## Test-retest reliability

Table 5 reports the means and SDs for all of the UK Biobank tests at Time 1 and Time 2 in a subsample of participants (n = 52) who completed the UK Biobank tests for a second time after a mean interval of 28.88 days (SD = 2.02). Generally, mean scores were higher at Time 2 than at Time 1; however, mean performance was significantly better at Time 2 compared to Time 1 only for UKB RT ($t(51) = 3.22$, $p = .002$), UKB Fluid IQ ($t(50) = 2.26$, $p = 0.03$), and UKB Symbol Digit ($t(51) = 3.37$, $p = .001$). The effect size for the difference between mean scores at Time 1 and Time 2 ranged from negligible (Cohen's $d = 0.00$) to moderate (Cohen's $d = 0.47$) and were highest for UKB Fluid IQ (Cohen's $d = 0.32$), UKB RT (Cohen's $d = 0.45$) and UKB Symbol Digit (Cohen's $d = 0.47$).

The test-retest reliabilities (Pearson and Spearman rank-order correlations) for each UK Biobank test are reported in Table 5. This table also contains some of the test-retest correlations reported elsewhere for some of the reference tests. UKB Pairs Matching had the lowest Pearson test-retest correlation ($r_{12} = 0.41$, $p = .003$). Test-retest reliability was high for UKB Picture Vocabulary ($r_{12} = 0.89$, $p < .001$) and UKB TMT part B ($r_{12} = 0.78$, $p < .001$). Test-

**Table 5. Test-retest[a] reliability for the UK Biobank cognitive tests (n = 52) and comparable reference tests.**

| UK Biobank tests | Time 1 | | Time 2 | | $p$ | Cohen's $d$ | Pearson $r_{12}$ | Spearman $rho_{12}$ | Reference tests | |
|---|---|---|---|---|---|---|---|---|---|---|
| | Mean | SD | Mean | SD | | | | | Reference test | Reference test $r_{12}$ |
| UKB Pairs Matching | 3.08 | 2.63 | 2.79 | 2.07 | 0.427 | 0.11 | 0.409** | 0.348* | WMS-IV Designs I[f] | 0.73 |
| | | | | | | | | | WMS-IV Designs II[f] | 0.72 |
| UKB Reaction time | 627.04 | 98.37 | 588.87 | 77.50 | 0.002 | 0.45 | 0.550*** | 0.585*** | DLRT Simple RT[g] DLRT Choice RT[g] | 0.64 0.83 |
| UKB Prospective Memory | 37[c] | 71.2%[d] | 48[c] | 92.3%[d] | - | 41 (78.8%)[e] | 0.453** | 0.453** | - | - |
| UKB Fluid IQ[b] | 6.80 | 2.05 | 7.39 | 2.14 | 0.028 | 0.32 | 0.607*** | 0.560*** | - | - |
| UKB Numeric Memory | 7.00 | 1.24 | 7.00 | 1.17 | 1.000 | 0.00 | 0.501*** | 0.517*** | WAIS-IV Digit Span[h] | 0.82 |
| UKB TMT part A | 246.85 | 90.59 | 231.98 | 72.22 | 0.211 | 0.18 | 0.478*** | 0.502*** | TMT part A[i] | 0.79 |
| UKB TMT part B | 475.52 | 137.51 | 466.42 | 151.98 | 0.506 | 0.09 | 0.775*** | 0.861*** | TMT part B[i] | 0.89 |
| UKB Symbol Digit | 18.85 | 5.63 | 21.12 | 4.91 | 0.001 | 0.47 | 0.583*** | 0.603*** | SDMT[j] | 0.80 |
| UKB Picture Vocabulary | 8.94 | 1.59 | 9.08 | 1.55 | 0.166 | 0.19 | 0.887*** | 0.828*** | NIH TB Picture Vocabulary[k] | ICC = 0.81 (95% CI 0.73 to 0.87) |
| | | | | | | | | | PPVT[l] | 0.94 |
| UKB PAL | 7.87 | 2.22 | 8.33 | 2.20 | 0.158 | 0.20 | 0.450** | 0.542*** | WMS-IV VPA I[f] | 0.79 |
| | | | | | | | | | WMS-IV VPA II[f] | 0.81 |
| UKB Tower Test | 10.48 | 3.36 | 11.10 | 3.07 | 0.202 | 0.18 | 0.434** | 0.410** | D-KEFS Tower test[m] | 0.44 |
| UKB Matrices | 9.02 | 1.79 | 9.40 | 1.84 | 0.153 | 0.20 | 0.445** | 0.442** | COGNITO Matrices[n] | 0.70 |

p, p-value from a paired-samples t-test comparing scores at Time 1 and Time 2; $r_{12}$, Pearson correlation between test scores at Time 1 and Time 2; $rho_{12}$, Spearman rank-order correlation between test scores at Time 1 and Time 2; UKB, UK Biobank; Fluid IQ, Fluid Intelligence; TMT, Trail Making Test; PAL, Paired Associate Learning; Matrices, Matrix Pattern Test; WMS-IV, Wechsler Memory Scale–Fourth Edition; DLRT, Deary-Liewald Reaction Time Test; RT, Reaction Time; WAIS-IV, Wechsler Adult Intelligence Scale–Fourth Edition; SDMT, Symbol Digit Modalities Test; NIH TB Picture Vocabulary, NIH Toolbox Picture Vocabulary Test; PPVT, Peabody Picture Vocabulary Test–Fourth Edition; VPA, Verbal Paired Associates; D-KEFS, Delis-Kaplan Executive Function System; ICC, Intraclass correlation.

*$p < .05$,

**$p < .01$,

***$p < .001$

[a]Test-retest interval = 28.88 (SD = 2.02) days.

[b]n = 51

[c]For UKB Prospective Memory, the value is the number of participants who were correct on the first attempt.

[d]For UKB Prospective Memory, the value is the percentage of participants who were correct on the first attempt.

[e]For UKB Prospective Memory, the value is n (percentage) agreement for whether participants gave the same response (i.e., correct or incorrect on first attempt) at Time 1 and Time 2.

[f]Test-retest interval mean = 23 days (range 14 to 84 days), n = 244 [10].

[g]Period-free reliability. Participants completed the DLRT Simple RT and DLRT Choice RT twice immediately one after the other, n = 20 [17].

[h]Test-retest interval mean = 22 days (range 8 to 82 days), n = 298 [32].

[i]Test-retest interval mean = 11 months, n = 384 [22].

[j]Test-retest interval mean = 29.40 days, n = 80 [11].

[k]Test-retest interval range 7 to 21 days, n = 89 [13].

[l]Test-retest interval mean = 30.1 days (range 14 to 48 days), n = 67 [21].

[m]Test-retest interval mean = 25 days (range 9 to 74 days), n = 101 [12].

[n]Test-retest interval range 2 to 3 weeks, n = 78 [23].

retest correlations for all other UK Biobank tests were moderate ($r_{12}$ = 0.43 to 0.61). The test-retest reliability found here for the UK Biobank tests tended to be lower than those reported elsewhere for the reference tests. For example, the test-retest reliability for WMS-IV VPA I and VPA II was $r_{12}$ = 0.79 and $r_{12}$ = 0.81, respectively [10], whereas the test-retest correlation

for UKB PAL was $r_{12} = 0.45$. The SDMT has been reported to have a test-retest reliability of 0.80 over a 30 day test-retest interval [11], whereas the test-retest reliability here for UKB Symbol Digit was found to be $r_{12} = 0.58$. Whereas the test-retest reliability reported here for the UKB Tower Test was relatively low ($r_{12} = 0.43$), this test-retest correlation was in line with that previously reported for the D-KEFS Tower Test ($r_{12} = 0.44$) [12].

## Discussion

Using a sample of 160 middle-aged and older adults this study investigated the concurrent validity and test-retest reliability of the UK Biobank cognitive tests. This study had three main findings: 1) generally, the UK Biobank tests correlated moderately-to-strongly with well-validated, standard tests designed to assess the same cognitive domain; 2) a measure of general cognitive ability can be created using all of the UK Biobank tests, as well as using only the five UK Biobank baseline tests, and these measures of general cognitive ability are highly correlated with a measure of general cognitive ability created using a battery of standard cognitive measures; 3) most of the UK Biobank tests showed moderate-to-high test-retest reliability, but these tended to be lower than those reported elsewhere for the reference tests.

### Concurrent validity

Despite the brief and non-standard nature of the UK Biobank cognitive tests, they tended to correlate moderately-to-strongly with well-validated cognitive tests that were designed to assess the same cognitive domain or specific ability. The UK Biobank cognitive tests mostly showed modest to good concurrent validity. Below, we summarise the findings from the concurrent validity analysis; however, when interpreting the concurrent validities, it is important to be aware that the degree of similarity between each of the UK Biobank tests and the chosen reference tests varies. Whereas some of the reference tests use the same items as the UK Biobank tests (e.g., NIH Toolbox Picture Vocabulary and UKB Picture Vocabulary), others are different versions of the same test (e.g., SDMT and UKB Symbol Digit), and others still are different tests that are thought to assess the same underlying cognitive ability (e.g., WMS-IV Designs and UKB Pairs Matching). Thus, some reference tests used here were better 'matches' for the UK Biobank tests than others, and therefore readers should bear this in mind when interpreting the respective UK Biobank-reference tests' associations.

The UK Biobank Picture Vocabulary test showed especially good concurrent validity. This test correlated very highly ($r = 0.83$) with the original version of this test—the NIH Toolbox Picture Vocabulary test—and also with another picture vocabulary test, the PPVT ($r = 0.74$). In a validation study of the NIH Toolbox [13], the correlation between the NIH Toolbox Picture Vocabulary test and the PPVT was $r = 0.78$, which is very similar to the correlation found here between the UK Biobank version of this test and the PPVT ($r = 0.74$). In addition, the UKB Picture Vocabulary test was also found to correlate highly ($r = 0.75$) with the NART, which is often used as an estimate of crystallised cognitive ability [14, 15]. The results from this study suggest that the UK Biobank Picture Vocabulary is a valid measure of crystallised ability and may be used as an estimate of premorbid cognitive functioning.

The UKB TMT part B and UKB Symbol Digit tests, which both correlated at greater than 0.6 with the original, paper-and-pencil versions of these tests [11, 16], also showed good concurrent validity. In addition to correlating highly with their reference tests, UKB TMT part A and UKB Symbol Digit also correlated positively with a number of other non-reference tests that also have a speeded component (e.g., DLRT Choice RT) providing additional support that these tests are assessing processing speed. Other UK Biobank tests which showed reasonably good concurrent validity (i.e., correlated relatively highly with the chosen reference test)

include the UKB RT, UKB Numeric Memory, UKB TMT part A, UKB PAL, and UKB Matrices. Of note, the UKB RT score, which is created from a mean of only 4 trials, correlated at 0.52 with DLRT Simple RT and at 0.43 with DLRT Choice RT, which are more detailed tests of reaction time created from a mean of 20 trials and 40 trials, respectively [17]. Therefore, despite the brief nature of the UKB RT test, it appears to have relatively good concurrent validity.

UKB Pairs Matching had only a moderate correlation ($r$ = -0.33) with WMS-IV Designs Total, the chosen reference test. The differences between UKB Pairs Matching and WMS-IV Designs may account for this lower correlation. Better performance on UKB Pairs Matching had stronger associations with better performance on D-KEFS Tower Test and COGNITO Matrices than it did with the chosen reference test. D-KEFS Tower Test and COGNITO Matrices are both visuospatial reasoning tests.

UKB Prospective Memory did not correlate highly with the chosen reference test ($r$ with RMBM Appointments = 0.22). Reasons for this low correlation could be that both UKB Prospective Memory and RMBM Appointments are very brief, 1–2 item tests, and a high proportion of participants scored full marks on these tests. For the one-item UKB Prospective Memory test, 69% of participants correctly answered this question correctly on the first attempt. For RMBM Appointments, 59% of participants scored 4/4. Therefore these tests had limited variance in the relatively healthy sample used here. Correctly answering the UKB Prospective Memory test correlated moderately with other memory tests (e.g., WMS-IV Designs and WMS-IV VPA), as well as tests of executive function (D-KEFS Tower Test) and reasoning (COGNITO Matrices).

Whereas the UKB Tower Test had moderate positive correlations with D-KEFS Tower Test ($r$ = 0.40)—the reference test—it had larger correlations with WMS-IV Designs Total, SDMT, and TMT part B. Like the UKB Tower Test, TMT part B is thought to measure executive function. The correlation with SDMT may reflect the fact that the UKB Tower Test was a timed test —participants were tasked with completing as many Tower trials as possible in 3 minutes— and may therefore be measuring processing speed as well as executive function. The WMS-IV Designs is a measure of visuospatial memory. The UKB Tower Test requires participants to mentally move the hoops on the pegs in their mind, therefore it is likely to also be measuring visuospatial abilities.

We did not include a reference test for UKB Fluid IQ, which was designed to assess fluid ability. A previous study using UK Biobank baseline data [18] found that scores on the UKB Fluid IQ test showed mean values that remained relatively stable between the ages of 40 and 60 years and therefore did not show the age-related decline across the adult lifespan that is the hallmark of fluid ability [2]. Hagenaars et al. [18] suggested that, because of the relative stability in middle-age, UKB Fluid IQ may in fact be measuring a more crystallised ability. In the present study, however, we found that UKB Fluid IQ was negatively correlated with age and that it correlated most strongly ($r \geq 0.38$) with tests of working memory (WAIS-IV Digit Span), and non-verbal reasoning (COGNITO Matrices)—tests thought to assess more fluid abilities [2]. Therefore, this test may be more fluid than was suggested by Hagenaars et al. [18], although it did also have moderate correlations with the three standard vocabulary tests.

The UKB PAL test exhibited negative skew (S1 File, S1 Fig) suggesting that most participants find this test quite easy. Despite the negative skew, scores on the UKB PAL test were found to correlate moderately with the M-ACE ($r$ = 0.47), a brief assessment of global cognitive functioning that is designed to identify individuals who may have possible cognitive impairment [19]. The UKB PAL test may be a useful test to identify individuals in UK Biobank who may have a possible cognitive impairment.

In addition to correlating relatively highly with the chosen reference tests, most UK Biobank cognitive tests also had positive correlations with many non-reference tests, and they

loaded strongly on the general cognitive component. When we write about tests correlating because they both assess the same 'cognitive domain' or 'underlying cognitive ability' it might also be in part or in whole because they both assess general cognitive ability (*g*). It is an error not to acknowledge this, as Schmidt [20] discusses in detail. However, mindful of the fact that there is variance beyond *g* and that is accounted for at the level of cognitive domains and specific abilities [9], and the fact that readers will wish to know how the largely-undocumented UK Biobank tests relate to better-validated tests, we think the references we have made to domains and specific abilities are appropriate.

## General cognitive ability

In the present study, we compared whether measures of general cognitive ability created using the brief, bespoke UK Biobank tests that were administered unsupervised correlated strongly with a measure of general cognitive ability created using well-validated tests administered under standardised conditions. The correlations between general cognitive ability created using well-validated tests and general cognitive ability created using the UK Biobank tests were high ($r$ = 0.83 for a measure created using all 11 UK Biobank tests; $r$ = 0.74 for the 5 baseline UK Biobank tests). These correlations reported here were lower than those reported in one study [5] that found that three measures of general cognitive ability created using three entirely different cognitive test batteries correlated nearly perfectly ($r \geq 0.99$). However, the correlations found here are in line with another study [6] that compared five different measures of general cognitive ability and found that they correlated at $r \geq 0.77$. This suggests that, despite the brief and non-standard nature of the UK Biobank cognitive assessment, a measure of general cognitive ability can be created using these tests. UKB TMT part B, UKB Matrices, UKB Numeric Memory, and UKB Fluid IQ all correlated at $\geq 0.55$ with the general measure of cognitive ability created using the standardised tests, suggesting these UK Biobank tests load strongly on general cognitive ability.

## Reliability

Test-retest correlations for UKB Picture Vocabulary ($r_{12}$ = 0.89) and UKB TMT part B ($r_{12}$ = 0.78) were high, and comparable to those reported for other, well-validated, measures of picture vocabulary (NIH Toolbox Picture Vocabulary intraclass correlation = 0.81; 95% CI 0.73 to 0.87 [13]; and PPVT $r_{12}$ = 0.94 [21]) and for the original paper-and-pencil version of the TMT part B ($r_{12}$ = 0.89 [22]). Therefore, UKB Picture Vocabulary and UKB TMT part B show good stability. Good short-term stability of cognitive tests is especially important when examining longitudinal change. Low stability means that any differences in scores over time may not be due to real change in test performance, but due to error of measurement.

Generally, the test-retest reliability for most of the UK Biobank tests was substantial. UKB RT, UKB Fluid IQ, UKB Numeric Memory, and UKB Symbol Digit had test-retest correlations of greater than 0.5. However, mean performance on the UKB Fluid IQ, UKB RT, and UKB Symbol Digit was found to be significantly higher at Time 2, compared to Time 1, suggesting these tests may be most prone to repeat testing effects. UKB Pairs Matching, UKB Prospective Memory, UKB TMT part A, UKB PAL, UKB Tower Test, and UKB Matrices had modest test-retest correlations (e.g., between 0.4 and 0.5). Although the test-retest correlations for the UK Biobank tests were found to be adequate, they tended to be lower than those reported previously for the reference tests, suggesting that the UK Biobank tests are less stable across time than well-validated tests administered under standardised conditions. The relative brevity of some of the UK Biobank tests might contribute to the lower reliability. However, UKB Matrices uses the same 15 items as COGNITO Matrices, and both are administered via a computer and yet the test-retest

correlation for UKB Matrices was $r_{12} = 0.44$, whereas the test-retest correlation reported for the COGNTIO Matrices test was higher, at $r_{12} = 0.70$ [23]. It is not clear why the UK Biobank tests have lower test-retest reliability than other measures of cognitive function.

Using the UK Biobank baseline and repeat data, Lyall et al. [3] investigated the stability of UKB Pairs Matching, UKB RT and UKB Fluid IQ. Like the current study, Lyall et al. [3] found that UKB Pairs Matching had the lowest test-retest reliability. However, the test-retest reliability for UKB Pairs Matching was substantially larger in the current study ($r_{12} = 0.41$) than was reported using UK Biobank data in Lyall et al. [3, 18], who reported the test-retest reliability of the UKB Pairs Matching test to be $r_{12} = 0.19$ [3]. The lower test-retest reliability reported in Lyall et al. [3] might be because they used a test-retest interval of over 4 years, which is much longer than the four-week test-retest interval used in the current study and therefore the test-retest correlation reported in Lyall et al. may in part reflect cognitive change over time, in addition to test stability. Despite the differences in the test-retest interval, the current study and the study by Lyall et al. [3] found very similar stability estimates for UKB RT (Lyall et al. $r_{12} = 0.54$ [3]; present study $r_{12} = 0.55$) and UKB Fluid IQ (Lyall et al. $r_{12} = 0.65$ [3]; present study $r_{12} = 0.61$), suggesting these tests do show relatively good stability.

## Other psychometric considerations in some UK Biobank cognitive tests

The UKB Numeric Memory test was designed as a backwards digit span task to assess working memory—the ability to temporarily store information in short-term memory long enough to manipulate it [15]. Backward digit span tasks require individuals to both remember a sequence of numbers and mentally reverse these numbers in their mind, and this differs from a forward digit span task where participants are only required to remember a sequence of digits [15]. Despite the fact that the UKB Numeric Memory test was designed to assess backward digit span, we found that only 14.2% of the sample tested in the current study reported performing a backward digit span. All other participants reported that they either carried out a forward digit span (72.3%), or they used a mixture of both techniques (13.5%). This means that, for the majority of participants, this test is not assessing the type of mental performance that it was intended to assess.

This study also found that nearly one-quarter of participants reported that they thought the test instructions for the UKB Tower Test were unclear. Given that the UK Biobank tests are administered unsupervised, and participants are expected to sit at a computer in a UK Biobank clinic and work through these tests independently, it is important that the test instructions are clear and the participant knows exactly what to do before starting the test proper. The UKB Tower Test had several pages of instructions. The length of the test instructions might be an important contributor to why participants reported that the test instructions for UKB Tower Test were unclear. Other UK Biobank tests with lengthy instructions, including UKB Symbol Digit (9.7%) and UKB TMT (8.3%) also had higher percentages of participants reporting that the test instructions for these tests were not clear, whereas tests with relatively short instructions, such as UKB Matrices (0.7%) and UKB Picture Vocabulary (1.4%) tended to have very few participants reporting that they thought the instructions were not clear. All of these tests, however, had practice examples which should allow participants to see what is involved before starting the task proper, even if they did not fully understand the test instructions before starting the practice trials.

## Advantages and limitations

The main advantage of this study is that the fully-automated UK Biobank cognitive assessment was compared to a large number of well-validated, standard cognitive tests that were

administered under standardised conditions. This meant that the brief and non-standard UK Biobank cognitive tests were compared to what many would consider to be the 'gold standard' measures of cognitive ability. For the current study, UK Biobank provided us with a stand-alone version of the UK Biobank cognitive assessment that is currently being administered at the UK Biobank imaging study. UK Biobank also provided us with a UK Biobank button-box to be used for the UKB RT test and with details about the computing equipment used at the UK Biobank clinic assessments which enabled us to very closely mimic the UK Biobank clinic cognitive assessment.

There are some limitations to the current study. The sample size is relatively small, especially for the test-retest sample. The testing conditions in the current study were not identical to the testing conditions used during the UK Biobank clinic assessments. In the current study, participants were assessed individually in a quiet room, free of distraction. The UK Biobank assessment centre could be busy and sometimes noisy (CF-R and IJD both spent a day at one UK Biobank testing centre during people's imaging visits). In the current study, the UK Biobank tests were administered in a more usual and standardised psychological testing environment. It is not clear whether the reliability and validity reported in the current study would differ if the UK Biobank tests had been administered in a busy and sometimes noisy environment that was seen when the authors visited the UK Biobank testing centre. In addition to the cognitive assessment administered at the UK Biobank assessment centre, UK Biobank have also collected cognitive data using web-based assessments. For the web-based assessment, participants are sent a link, via email, and were to complete the cognitive tests at home. The testing conditions of the web-based assessment, therefore, were even less controlled than at the UK Biobank assessment centre. We do not know whether the results of the current study would generalise to the UK Biobank web-based assessment.

This study only examined some aspects of the validity and reliability of the UK Biobank tests. We did not examine, for example, their internal consistency or predictive validity for other 'real-world' outcomes. Another limitation is that the sample used in the current study was relatively highly educated. The mean years of full-time education in the current sample was 16.19 years. However, UK Biobank participants—especially repeat study samples—were also highly educated. At baseline, 18% of participants reported having a college or university degree. Data collection for the imaging study is still ongoing; however, almost half (48%) of participants who have attended so far report having a college or university degree (http://biobank.ndph.ox.ac.uk/showcase/field.cgi?id=6138). Because the samples used here (and in UK Biobank) mostly consists of relatively highly educated individuals, it is likely that the range of cognitive test scores found here are not representative of the range of cognitive test scores that would be identified in the entire population. Therefore, the correlations reported here between the UK Biobank tests and the reference tests may be attenuated, compared to those reported if we had used a samples more representative of the general population.

## Conclusions

This study examined the concurrent validity and test-retest reliability of the enhanced UK Biobank cognitive assessment that is currently being administered to UK Biobank participants attending the UK Biobank imaging study. The UK Biobank cognitive tests are administered using a fully-automated touch-screen assessment, and participants complete these tests unsupervised. The tests in UK Biobank tend to be short. They were created specifically for UK Biobank, or were adapted for use in a fully-automated assessment. The present study found that they showed a range of concurrent validity coefficients with well-validated, standard tests of cognitive ability, and most tests tended to have moderate-to-good test-retest reliability. UK

Biobank is one of the largest and most detailed health resources available worldwide. This paper provides currently-lacking information on the psychometric properties of the UK Biobank cognitive tests. Researchers wishing to use the UK Biobank cognitive data should consider analysing cognitive test data from those tests which have been found here to have both moderate-to-high concurrent validity and short-term stability.

## Supporting information

**S1 File. Supplementary materials for Reliability and validity of the UK Biobank cognitive tests.**
(PDF)

**S1 Table. Pearson correlations (below the diagonal) and age-adjusted Pearson correlations (above the diagonal) between all cognitive tests and demographic variables.**
(XLSX)

**S2 Table. Spearman rank-order correlations (below the diagonal) and age-adjusted Spearman rank-order correlations (above the diagonal) between all cognitive tests and demographic variables.**
(XLSX)

**S1 Data. Study data.**
(CSV)

## Acknowledgments

We thank the participants of the present study. We thank UK Biobank for providing us with a stand-alone version of the UK Biobank cognitive assessment, a UK Biobank button-box, and advice on UK Biobank computer equipment. We especially thank Alan Young, Keith Anderson and other members of the NDPH Core Programming and Testing teams for producing a stand-alone version of the UK Biobank cognitive tests. We thank Michelle Miranthi for her assistance with the data entry. We thank John Gallacher, Marcus Richards, and our late colleague John Starr, who were members of Work Package 10 of Dementias Platform UK (DPUK) alongside IJD and CF-R.

Join Dementia Research was used to identify potential participants for this study.

## Author Contributions

**Conceptualization:** Chloe Fawns-Ritchie, Ian J. Deary.

**Data curation:** Chloe Fawns-Ritchie.

**Formal analysis:** Chloe Fawns-Ritchie.

**Investigation:** Chloe Fawns-Ritchie.

**Methodology:** Chloe Fawns-Ritchie.

**Project administration:** Ian J. Deary.

**Resources:** Chloe Fawns-Ritchie.

**Supervision:** Ian J. Deary.

**Writing – original draft:** Chloe Fawns-Ritchie.

**Writing – review & editing:** Chloe Fawns-Ritchie, Ian J. Deary.

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
