## [Decision Letter · Decision Letter 0]

23 Dec 2019

PONE-D-19-30264

Reliability and validity of the UK Biobank cognitive tests

PLOS ONE

Dear Ms Fawns-Ritchie,

Thank you for submitting your manuscript to PLOS ONE. After careful consideration, we feel that it has merit but does not fully meet PLOS ONE’s publication criteria as it currently stands. Therefore, we invite you to submit a revised version of the manuscript that addresses the points raised during the review process.

Please, see the comments of three Reviewers appended at the bottom of this letter. Because this might be considered as a major review, please notice that a resubmission will require an additional round of reviews, and that the final outcome of the process cannot be predicted at this point. If you decide to resubmit a revised version of your manuscript, please provide either a proper answer or rebuttal to each of the suggestions that were raised by the Reviewers.

We would appreciate receiving your revised manuscript by Feb 04 2020 11:59PM. To enhance the reproducibility of your results, we recommend that if applicable you deposit your laboratory protocols in protocols.io, where a protocol can be assigned its own identifier (DOI) such that it can be cited independently in the future. For instructions see: http://journals.plos.org/plosone/s/submission-guidelines#loc-laboratory-protocols

We look forward to receiving your revised manuscript.

Kind regards,

Angel Blanch, Ph.D.

Academic Editor

PLOS ONE

Journal Requirements:

Reviewers' comments:

Reviewer's Responses to Questions

**Comments to the Author**

1. Is the manuscript technically sound, and do the data support the conclusions?

Reviewer #1: Yes

Reviewer #2: Yes

Reviewer #3: Yes

2. Has the statistical analysis been performed appropriately and rigorously? 

Reviewer #1: N/A

Reviewer #2: Yes

Reviewer #3: I Don't Know

3. Have the authors made all data underlying the findings in their manuscript fully available?

Reviewer #1: No

Reviewer #2: No

Reviewer #3: No

4. Is the manuscript presented in an intelligible fashion and written in standard English?

Reviewer #1: Yes

Reviewer #2: Yes

Reviewer #3: Yes

5. Review Comments to the Author

Reviewer #1: a very similar attempt to validate the Biobank tests by the same author was published online in july 2019, see

https://www.medrxiv.org/content/medrxiv/early/2019/07/15/19002204.full.pdf . Would have been appropiate a "Validation of UK Biobank tests" with coherent total results. However, as the manuscript was earlier published elsewhere, it must be rejected.

Reviewer #2: The authors present a study of the reliability and validity of the cognitive tests of the UK Biobank study.

Motivated by a discrepancy between the widespread use of these tests and the limited psychometric information about them, they conduct an investigation of the relations of the UK Biobank tests to a set of reference tests as well as a 4-week retest study in an independent, small-to-medium-sized sample of elderly adults.

I can see no fundamental issues with the design and conduct of the study, and the manuscript is, in general, well written. I therefore recommend acceptance of this manuscript, provided that some minor issues, as detailed below, are resolved.

Abstract

1. The introduction states that this study has three aims: examination of the concurrent validity of the tests, extraction and comparison of a factor of general cognitive ability from both the UK Biobank and reference tests, and investigation of test-retest reliability. The abstract, however, speaks of only two aims: investigation of concurrent validity and test-retest reliability. This discrepancy should be resolved.

Introduction

2. The introduction is, in parts, relatively lengthy and contains information that is redundant and/or not necessary to understand the remainder of the manuscript. Examples are, in my eyes, lines 45-46, 50-52, 59-64, 77-79. I recommend to shorten the introduction to what is necessary for understanding of the background and rationale of the current study.

3. The authors mention that several large-scale ‘repeat studies’ have been conducted within the UK Biobank study (ll. 52 ff.). This is highly relevant for the current research. Here, my questions are whether or not the data of these repeat studies are available and suitable for reliability analyses. If the Lyall et al. study is based on these data, it should already be mentioned here.

4. The section about the three aims of the study, from line 106 to 148, is rather lengthy and unbalanced. For example, the section on aim 2 (lines 117-141) is given considerably more space than aims 1 and 3, and contains background information that should be moved to earlier parts of the introduction. Also, lines 143-148 reflect background information. I recommend that the introduction ends with a relatively concise statement of the research aims (and hypotheses, where appropriate), and that these aims follow from the preceding sections of the introduction. At least for aim 2, which presents a new idea that does not follow from the abstract or preceding introduction, this is not yet the case.

Methods

5. Lines 181-183 provide background info on the utility of cognitive screening tests. This could be moved to the introduction, but is not required in the methods section.

6. Lines 193-199 constitute a critical appraisal of the similarity between the UK Biobank tests and the chosen reference tests. This should have a place in the discussion section rather than the methods section.

7. Please comment on whether or not range restriction was observed for any tests, and if so, about potential consequences for reliability and validity.

8. I may have missed it, but please explain why several tests were omitted from the construction of general cognitive ability factor (TMT A and RMBM appointments from the reference tests; UKB TMT A from the UK Biobank tests).

9. Besides reporting their overall correlation, please include a comment on the similarity/equivalence of the factor structure of the g:UKB-11 and g:reference-11 factors, including the discrepancy of loadings for the UKB Picture Vocabulary Test vs. the NIH Picture Vocabulary Test. Also, what is the interpretation of the second factor that has been indicated by the Scree plots?

Results

10. The correlations of the UK Biobank tests with the sets of other tests are reported in separate sections. However, several subsections of the the “Associations with reference tests” section also contain correlations of the UKB Biobank tests with the M-ACE tests, which is a ‘general test’ according to the authors’ terminology. Therefore, the M-ACE results should be removed from this section.

11. Some consistency is required with regard to when specific correlations are reported in detail (i.e, exact numerical value), in a summary fashion, or not at all in the text of the results section. Currently it appears to me that correlations with the reference tests are always reported, plus a selection of other tests, depending on whether or not these are significant, conceptually related, and/or exceed a certain threshold. It is certainly not possible/meaningful to describe each and every correlation within the text, but the criteria for highlighting specific correlations should be consistent and transparent. Regarding the UKB TMT A test, for example, not only the correlation with the reference test (TMT A), the TMT B, and the SDMT tests should be reported in the text, but also the correlations with the WMS-IV, DLRT choice RT, COGNITO tests – because these are significant as well. Please also check for other tests.

12. Please check the correlation of UKB TMT B with DLRT choice RT: it is 0.54 in the manuscript text (line 420), but 0.514 in Table 1.

13. Lines 440-446 reflect, in my eyes, an interpretation of the reported findings, and this should be moved from the results section to the discussion.

14. The description and results of the sensitivity analysis (ll. 479-480) could be moved to the supplementary material entirely, to save some space in the results section.

15. As a suggestion, a table of the correlations between ‘g:reference-11’ and the UK Biobank tests (ll. 489 ff.) and between ‘g:UKB-11’ and the reference tests could be moved from the supplementary material to the results section, because this will give a more complete picture how the two measures of ‘g’ are reflected in the complementary tests, respectively.

16. Please do not describe non-significant results (unless equivalence tests were conducted) as indications of no relation or difference (e.g. ll. 546f.): “did not reveal differences” is more appropriate, in my eyes, than “revealed that XYZ did not differ”.

17. I have some trouble understanding the values presented for UKB Prospective Memory in Table 3, especially why the mean is given as a an absolute value and the SD as a percent value. Could this be a formatting error?

Discussion

18. For the sake of completeness, I would suggest to (at least briefly) mention each UK Biobank in the ‘concurrent validity’ and ‘reliability’ sections of the discussion, as the reader may want to look up specific tests of interest. Currently, the criterion which tests are highlighted here is not very clear to me, maybe just those with relatively high or low reliabilites/validites? And as far as I can see, only about half to the UK biobank tests (6/12) are discussed in the validity section, and 4/12 in the reliability section.

19. The low correlation of the UKB Prospective Memory test with its reference test needs further investigation.

20. I am not really sure what to make of the comparison of the reliabilities with the previous study by Lyall et al. (ll. 663). Please comment on the partially agreeing, partially non-agreeing results.

21. The beginning of the ‘concurrent validity’ section is a little bit awkward, because it focuses on a very specific caveat, which should have its part in the later parts of the discussion.

22. The sample demographics of the current study are not fully comparable to those of the UK Biobank. Specifically, the sample of the present study is older. This could impact the observed reliabilities and validities, and hence should be made clear in the limitations section.

23. There are other forms of validity than concurrent validity, e.g. correlations with measures other than psychometric ones, or prediction of ‘real-world’ criteria. This should be acknowledged in the limitations section.

Reviewer #3: In this paper, authors aim to analyze psychometric properties of a widely used health resource which contains cognitive function data known as UK Biobank. In their study, authors developed and performed a well design that showed good concurrent validity and test-retest reliability. Moreover, autohors found high correlation between a g component created using the UK Biobank tests and a a g component created using reference tests. Nevertheless, data unavailability non accomplish PLOS Data policy.

Nevertheless, I have some questions regarding the paper:

1. As it is stated in the manuscript, the second aim of the study was to investigate wether a component of general cognitive ability (g) was present in the correlations among the unsupervised UK Biobank tests, and wether any such g component correlated highly with a measure of general cognitive ability created using the reference tests administered by a trained tester under standardised conditions. I would like to know why information related to this aim (results and conclusions) was not included in abstract nor conclusions section

2. Was there any schedule for test-administration? If yes, was this counter-balanced? It has been shown that circadian typology can influence the results of cognitive tests depending on the time at which these tests are administered (synchronicity effect).

6. PLOS authors have the option to publish the peer review history of their article (what does this mean?). If published, this will include your full peer review and any attached files.

Reviewer #1: No

Reviewer #2: No

Reviewer #3: No

---

## [Author Response · Author response to Decision Letter 0]

27 Feb 2020

Response to Reviewers

Reviewer 1

1. a very similar attempt to validate the Biobank tests by the same author was published online in july 2019, see

https://www.medrxiv.org/content/medrxiv/early/2019/07/15/19002204.full.pdf . Would have been appropiate a "Validation of UK Biobank tests" with coherent total results. However, as the manuscript was earlier published elsewhere, it must be rejected.

Response: The article you have linked to (i.e., https://www.medrxiv.org/content/medrxiv/early/2019/07/15/19002204.full.pdf) is a preprint version of the paper. A preprint, which is not peer-reviewed, involves posting a manuscript to a public server before the peer review process has taken place. Preprints are used to quickly circulate the results of the paper to other researchers as the peer review process can sometimes be a length process. 

As stated on the PLOS ONE website, “PLOS encourages authors to post preprints as a way to accelerate the dissemination of research” (see https://journals.plos.org/plosone/s/preprints). 

Reviewer 2

1. The introduction states that this study has three aims: examination of the concurrent validity of the tests, extraction and comparison of a factor of general cognitive ability from both the UK Biobank and reference tests, and investigation of test-retest reliability. The abstract, however, speaks of only two aims: investigation of concurrent validity and test-retest reliability. This discrepancy should be resolved.

Response: We thank Reviewer 2 for their helpful and detailed comments. This study does have three main aims, and we have updated the abstract accordingly. In addition to detailing the concurrent validity and the test-retest reliability of the UK Biobank tests, we have now included in the abstract that we investigated whether a general measure of cognitive ability could be extract from the UK Biobank cognitive tests, and examined the correlation between this measure of cognitive ability and a measure of general cognitive ability created using well-validated cognitive tests. We have added the following information to the abstract: 

“Two measures of general cognitive ability were created by entering scores on the UK Biobank cognitive tests, and scores on the reference tests, respectively, into separate principal component analyses and saving scores on the first principal component.” (Page 2, lines 23-25)

“The measure of general cognitive ability based on the UK Biobank cognitive tests correlated at r=0.83 (p<.001) with a measure of general cognitive ability created using the reference tests.” (Page 2, lines 28-30)

2. The introduction is, in parts, relatively lengthy and contains information that is redundant and/or not necessary to understand the remainder of the manuscript. Examples are, in my eyes, lines 45-46, 50-52, 59-64, 77-79. I recommend to shorten the introduction to what is necessary for understanding of the background and rationale of the current study.

Response: We have removed the suggested sections of text. One exception is that we have kept information relating to the other data that is collected in UK Biobank. We think it is important for the reader to be aware of the size and the breadth of data collected in UK Biobank, and that the cognitive data collected was only one small part of this study. We have, however, reduced to length of this section:

“…completed a touchscreen questionnaire collecting information on health and lifestyle. Physical measurements and biological samples were also collected during this clinic visit.” (Page 3, lines 39-40)

3. The authors mention that several large-scale ‘repeat studies’ have been conducted within the UK Biobank study (ll. 52 ff.). This is highly relevant for the current research. Here, my questions are whether or not the data of these repeat studies are available and suitable for reliability analyses. If the Lyall et al. study is based on these data, it should already be mentioned here.

Response: Alongside the baseline UKB Biobank data, the data from the UK Biobank repeat clinic visits are available for researchers to use in their research. The first paragraph of the introduction is to show the reader what cognitive data has been collected in UK Biobank. For clarity, we have removed the term “repeat studies” as the reader may think we are referring to published papers using the repeat cognitive data. Instead, we wanted readers to be aware that UK Biobank has repeat cognitive data available for researchers to use. We have changed the introduction to the following:

“Subsamples of UK Biobank participants have undergone repeat testing. Between 2009 and 2013, 20,000 UK Biobank participants returned and completed the baseline assessment again.” (Page 3, lines 40-42) 

The Lyall et al. study did use the UK Biobank baseline and repeat data to examine the test-retest reliability of some of the cognitive tests administered at baseline. When describing the Lyall et al. study, we have now made it explicitly clear that this study uses baseline and repeat UK Biobank data to calculate the test-retest reliability of some of the cognitive tests administered at baseline: 

“Using the UK Biobank data collected at baseline and the repeat visit, one study [3] examined the stability (test-retest reliability) of the original UK Biobank tests over a four year interval.” (Page 4, lines 81-83)

We emphasise that Lyall et al. did not study all of the cognitive tests that are now available on the UK Biobank sample.

4. The section about the three aims of the study, from line 106 to 148, is rather lengthy and unbalanced. For example, the section on aim 2 (lines 117-141) is given considerably more space than aims 1 and 3, and contains background information that should be moved to earlier parts of the introduction. Also, lines 143-148 reflect background information. I recommend that the introduction ends with a relatively concise statement of the research aims (and hypotheses, where appropriate), and that these aims follow from the preceding sections of the introduction. At least for aim 2, which presents a new idea that does not follow from the abstract or preceding introduction, this is not yet the case.

Response: As suggested, we have now added this background information to earlier sections of the introduction. For example, we now have a new paragraph in the introduction detailing previous research on the creation of a general cognitive ability component (page 5, lines 91-104). We have moved information about typical test-retest intervals into the paragraph describing the test-retest reliability of the UK Biobank baseline tests (page 5, lines 86-87). This restructuring of the introduction means that the analyses and hypotheses are now summarised in one concise paragraph: 

“To investigate the reliability and validity of the UK Biobank tests, three sets of analyses were carried out. First, the concurrent validity of the UK Biobank tests was investigated by correlating scores on the UK Biobank tests with scores on the reference tests. We predicted that the correlations between UK Biobank tests and reference tests that assessed a similar cognitive domain should show higher correlations than those between UK Biobank tests and reference tests that assessed different cognitive domains. Second, we investigated whether a g component was present in the correlations among the unsupervised UK Biobank cognitive tests, and tested whether any such g component correlated highly with a g component created using the reference tests which were administered by a trained tester under standardised conditions. The present study predicted that the correlation between a g component created using the UK Biobank tests and a g component created using the reference tests would be high (e.g., r > 0.7). Third, we characterised the short-term test-retest stability of the UK Biobank tests by correlating the cognitive test scores from time 1 and time 2 (approximately 4 weeks apart).” (Page 6, lines 119-131)

5. Lines 181-183 provide background info on the utility of cognitive screening tests. This could be moved to the introduction, but is not required in the methods section.

Response: We have moved information on the use of screening tests as a way for identifying possible cognitive impairment from the methods section to the introduction: 

“Participants also completed brief screening tests of cognitive impairment and measures of subjective memory complaints, which are often applied in studies of normal and pathological ageing to measure global cognitive functioning (hereinafter we will call these ‘general’ tests).” (Page 6, lines 113-116)

6. Lines 193-199 constitute a critical appraisal of the similarity between the UK Biobank tests and the chosen reference tests. This should have a place in the discussion section rather than the methods section.

Response: The section making the reader aware that some of the chosen reference tests are better matches for the UK Biobank cognitive tests than others has now been moved from the methods section to the discussion:

“Below, we summarise the findings from the concurrent validity analysis; however, when interpreting the concurrent validities, it is important to be aware that the degree of similarity between each of the UK Biobank tests and the chosen reference tests varies. Whereas some of the reference tests use the same items as the UK Biobank tests (e.g., NIH Toolbox Picture Vocabulary and UKB Picture Vocabulary), others are different versions of the same test (e.g., SDMT and UKB Symbol Digit), and others still are different tests that are thought to assess the same underlying cognitive ability (e.g., WMS-IV Designs and UKB Pairs Matching). Thus, some reference tests used here were better ‘matches’ for the UK Biobank tests than others, and therefore readers should bear this in mind when interpreting the respective UK Biobank-reference tests’ associations.” (Page 26, lines 585-593)

7. Please comment on whether or not range restriction was observed for any tests, and if so, about potential consequences for reliability and validity.

Response: The sample used in the current study are relatively highly educated. Therefore this sample is not representative of the population as a whole. Education and cognitive ability are highly correlated and therefore the cognitive test scores reported in this paper are unlikely to be representative of the general population. We have updated the discussion section of the paper to make readers aware that, because the cognitive test scores are likely to be restricted in range, then the correlations reported here are likely to be attenuated compared to the correlations if we had used a more representative sample:

“Because the samples used here (and in UK Biobank) mostly consists of relatively highly educated individuals, it is likely that the range of cognitive test scores found here are not representative of the range of cognitive test scores that would be identified in the entire population. Therefore, the correlations reported here between the UK Biobank tests and the reference tests may be attenuated, compared to those reported if we had used a samples more representative of the general population.” (Page 34, lines 776-781)

8. I may have missed it, but please explain why several tests were omitted from the construction of general cognitive ability factor (TMT A and RMBM appointments from the reference tests; UKB TMT A from the UK Biobank tests).

Response: We wanted to create a measure of general cognitive ability with the reference tests so that we have a measure of g created using a large number of diverse (with respect to the cognitive domains being assessed) standard cognitive tests. We wanted to create this measure so that we could compare it to a measure of general cognitive ability created with the brief and non-standard UK Biobank tests. We did not include the RMBM Appointments into the measure of general cognitive ability created with the reference tests because it is a very brief, two item test with limited variability. 

“This measure of general cognitive ability was designed to reflect a g component created using well-validated and comprehensive cognitive tests that have been viewed as the ‘gold standard’ cognitive measures. As the RMBM Appointments test is brief and contains only 2 items, this test was not included in this comprehensive measure of general cognitive ability.” (Page 13, lines 278-282)

Tests such as TMT and the DLRT consist of multiple highly correlated parts. We only included one score from each test to make sure these highly-correlated parts of tests did not overly influence the overall general cognitive ability score. We entered part B of the TMT as the TMT is often used to assess executive function and part B is the more “executive” part as it is assessing switching ability, whereas part A is often assumed to assess more processing speed ability. We used Choice RT from the DLRT because Choice RT is a more cognitively demanding task than Simple RT. We have updated the methods section to inform the reader why these specific tests were not included in our measure of general cognitive ability: 

“Only one score from each cognitive test was used to create each measure of general cognitive ability to ensure that highly correlated parts of tests do not overly influence the general cognitive ability score. For DLRT, the Choice RT part was chosen as this is a more cognitively challenging task than Simple RT. For TMT and UKB TMT, part B was used because TMT is thought to be a test of executive function—specifically switching ability—and part B is thought to assess this switching ability, whereas part A is often thought to be assessing processing speed.” (Page 12, lines 260-266)

9. Besides reporting their overall correlation, please include a comment on the similarity/equivalence of the factor structure of the g:UKB-11 and g:reference-11 factors, including the discrepancy of loadings for the UKB Picture Vocabulary Test vs. the NIH Picture Vocabulary Test. Also, what is the interpretation of the second factor that has been indicated by the Scree plots?

Response: We have now included sections on the interpretation of the PCA results, and include information on the similarity and differences between the factor structure of the g:UKB-11 and g:reference-11. We have also added an interpretation of the second factor indicated by the Scree plots and why we think there is a discrepancy of the loadings for the UKB picture Vocabulary Test and the NIH Toolbox Picture Vocabulary Test:

“Inspection of the rotated loadings suggests that the first component appears to reflect processing speed. The tests which load most highly on this component include TMT part B (-0.83), SDMT (0.82), WAIS-IV Designs Total (0.73), and DLRT Choice RT (-0.71). Non-speeded verbal tests load highly on the second component. The loadings for the NART, NIH Toolbox Picture Vocabulary, and PPVT were 0.91, 0.90, and 0.84, respectively.” (Page 13, lines 285-289)

“Like the results from the PCA for the reference tests, the first rotated component from the UKB tests appears to reflect processing speed, whereas the second component reflects non-speeded and verbal abilities. Examining the rotated loadings, tests that load highly on the first component include UKB TMT part B (-0.82), UKB Symbol Digit (0.80), and UKB Tower Test (0.71). UKB Picture Vocabulary loads highly on the second component (0.91). UKB PAL (0.48) and UKB Fluid IQ (0.45)—two verbal tests—also load moderately highly on the second component.” (Page 14, lines 299-304)

“The results of the PCA of 11 reference tests, and the PCA of 11 UK Biobank tests were generally similar, with the first rotated component reflecting processing speed and the second reflecting non-speeded, verbal ability. The first unrotated principal component (i.e., g) also accounted for a similar proportion of the variance in the test scores (35% and 34% for the reference and the UK Biobank tests, respectively). As a result of the fact that many of the tests used to create these g components were speeded tests, these measures of general cognitive ability we have created are largely measuring speeded/fluid cognitive abilities. We also note that only one vocabulary test was used to create g:UKB-11, whereas three were used in the creation of g:reference-11. This discrepancy is likely to be an important reason why the loading on the first unrotated principal component for UKB Picture Vocabulary (0.19) is lower than for NIH Toolbox Picture Vocabulary (0.51).” (Page 14, lines 307-316)

10. The correlations of the UK Biobank tests with the sets of other tests are reported in separate sections. However, several subsections of the the “Associations with reference tests” section also contain correlations of the UKB Biobank tests with the M-ACE tests, which is a ‘general test’ according to the authors’ terminology. Therefore, the M-ACE results should be removed from this section.

Response: We have now removed all reference of the M-ACE from the section “Associations with reference tests”.

11. Some consistency is required with regard to when specific correlations are reported in detail (i.e, exact numerical value), in a summary fashion, or not at all in the text of the results section. Currently it appears to me that correlations with the reference tests are always reported, plus a selection of other tests, depending on whether or not these are significant, conceptually related, and/or exceed a certain threshold. It is certainly not possible/meaningful to describe each and every correlation within the text, but the criteria for highlighting specific correlations should be consistent and transparent. Regarding the UKB TMT A test, for example, not only the correlation with the reference test (TMT A), the TMT B, and the SDMT tests should be reported in the text, but also the correlations with the WMS-IV, DLRT choice RT, COGNITO tests – because these are significant as well. Please also check for other tests.

Response: In the results section, we have now explicitly stated that throughout the “Associations with reference tests” section we first report the correlation between the UK Biobank test and the test(s) chosen as the reference test, and then report all correlations with the UK Biobank test that are greater than 0.3, which would reflect a moderate association between the UK Biobank test and standard tests. 

“In going through each UK Biobank cognitive test’s results below we first describe the correlation with the respective reference test(s), and then we highlight correlations with ‘non-reference’ tests that have absolute effect sizes greater than 0.3.” (Page 18, lines 394-397)

For each UK Biobank test, effect sizes of > 0.3 are now reported. For example: 

“UKB TMT part A. UKB TMT part A correlated positively with TMT part A (r = 0.44, p < .001)—the chosen reference test—and TMT part B (r = 0.46, p < .001). Faster completion of UKB TMT part A also moderately correlated with better performance on the SDMT (r = -0.47), DLRT Choice RT (r = 0.43), WMS-IV Designs Total (r = -0.39), and COGNITO Matrices (r = -0.35) (for all, p < .001).” (Page 19, lines 427-430)

12. Please check the correlation of UKB TMT B with DLRT choice RT: it is 0.54 in the manuscript text (line 420), but 0.514 in Table 1.

Response: Thank you for highlighting this typo. We have now changed the correlation in the text to the correct value (r = -0.51).

“Being quicker on UKB TMT part B was also moderately correlated with better performance on SDMT (r = -0.54), DLRT Choice RT (r = 0.51)…” (Page 19, lines 433-434)

13. Lines 440-446 reflect, in my eyes, an interpretation of the reported findings, and this should be moved from the results section to the discussion.

Response: These lines have been removed from the results section, and an interpretation of the correlations between UKB Pairs Matching and the reference tests is now provided in the discussion section:

“UKB Pairs Matching had only a moderate correlation (r = -0.33) with WMS-IV Designs Total, the chosen reference test. The differences between UKB Pairs Matching and WMS-IV Designs may account for this lower correlation. Better performance on UKB Pairs Matching had stronger associations with better performance on D-KEFS Tower Test and COGNITO Matrices than it did with the chosen reference test. D-KEFS Tower Test and COGNITO Matrices are both visuospatial reasoning tests.” (Page 27, lines 616-621)

We have similarly removed interpretations of the correlations for other UK Biobank tests from the results section, and moved these to the discussion.

14. The description and results of the sensitivity analysis (ll. 479-480) could be moved to the supplementary material entirely, to save some space in the results section.

Response: The reporting of the results of the sensitivity analysis has been moved to the supplementary materials (S1 File, Supplementary Results). A brief description of these results is now provided in the main text:

“The correlation was similar when re-run using a measure of general cognitive ability that was created excluding scores on the COGNITO Matrices and NIH Toolbox Picture Vocabulary test, which share items with UKB Matrices and UKB Picture Vocabulary (see S1 File, Supplementary Results).” (Page 21, lines 483-486)

15. As a suggestion, a table of the correlations between ‘g:reference-11’ and the UK Biobank tests (ll. 489 ff.) and between ‘g:UKB-11’ and the reference tests could be moved from the supplementary material to the results section, because this will give a more complete picture how the two measures of ‘g’ are reflected in the complementary tests, respectively.

Response: Tables of the correlations between g:reference-11 with the UK Biobank tests (Table 3) and between g:UKB-11 and g:UKB-5 with the reference tests (Table 4) have now been moved from the supplementary materials to the main document. 

16. Please do not describe non-significant results (unless equivalence tests were conducted) as indications of no relation or difference (e.g. ll. 546f.): “did not reveal differences” is more appropriate, in my eyes, than “revealed that XYZ did not differ”.

Response: We have now changed the wording of this section to the following: 

“A between-group ANOVA did not reveal any differences in UKB Numeric Memory scores by technique used (F (2, 136) = 0.518, p = .602).” (Page 24, lines 544-546)

17. I have some trouble understanding the values presented for UKB Prospective Memory in Table 3, especially why the mean is given as a an absolute value and the SD as a percent value. Could this be a formatting error?

Response: The values for UKB Prospective Memory in Table 5 (previously Table 3) are the number and percentage of participants who correctly answered the UKB Prospective Memory question on the first attempt. A footnote explaining the values for UKB Prospective Memory was missing from this table. Thank you for highlighting this omission. A footnote has now been added to the bottom of this table to make it clear to readers that the values for the UKB Prospective Memory test are the number and percentage of participants who answered this question correctly on the first attempt. 

18. For the sake of completeness, I would suggest to (at least briefly) mention each UK Biobank in the ‘concurrent validity’ and ‘reliability’ sections of the discussion, as the reader may want to look up specific tests of interest. Currently, the criterion which tests are highlighted here is not very clear to me, maybe just those with relatively high or low reliabilites/validites? And as far as I can see, only about half to the UK biobank tests (6/12) are discussed in the validity section, and 4/12 in the reliability section.

Response: The tests which were mentioned in the discussion tended to be those which had especially high or low reliabilities or validities, or that showed an unexpected pattern of association. Most of the tests which were previously omitted showed the expected pattern of correlations. To insure that these previously omitted tests are mentioned in the discussion, we have added sections to detail that many of the tests showed reasonably good concurrent validity and test-retest reliability:

“Other UK Biobank tests which showed reasonably good concurrent validity (i.e., correlated relatively highly with the chosen reference test) include the UKB RT, UKB Numeric Memory, UKB TMT part A, UKB PAL, and UKB Matrices.” (Page 27, lines 609-611)

“Generally, the test-retest reliability for most of the UK Biobank tests was substantial. UKB RT, UKB Fluid IQ, UKB Numeric Memory, and UKB Symbol Digit had test-retest correlations of greater than 0.5.” (Page 30, lines 690-692)

“UKB Pairs Matching, UKB Prospective Memory, UKB TMT part A, UKB PAL, UKB Tower Test, and UKB Matrices had modest test-retest correlations (e.g., between 0.4 and 0.5).” (Page 30, lines 694-695)

19. The low correlation of the UKB Prospective Memory test with its reference test needs further investigation.

Response: The low correlation between the UKB Prospective Memory test and the RMBM Appointments test was surprising. One possible reason for such a low correlation is because the UKB Prospective Memory test consists of only one item, and the RMBM Appointments test consists of only two items. In addition, many participants scored full marks on these tests, therefore they have limited variance in this healthy population. We have updated the discussion to highlight to the reader the possible reasons for this low correlation: 

“UKB Prospective Memory did not correlate highly with the chosen reference test (r with RMBM Appointments = 0.22). Reasons for this low correlation could be that both UKB Prospective Memory and RMBM Appointments are very brief, 1-2 item tests, and a high proportion of participants scored full marks on these tests. For the one-item UKB Prospective Memory test, 69% of participants correctly answered this question correctly on the first attempt. For RMBM Appointments, 59% of participants scored 4/4.” (Pages 27-28, lines 622-627).

20. I am not really sure what to make of the comparison of the reliabilities with the previous study by Lyall et al. (ll. 663). Please comment on the partially agreeing, partially non-agreeing results.

Response: Despite the differences in the length of the test-retest interval in the current study and the study by Lyall et al., both showed very similar results for the reliability of UKB RT and UKB Fluid IQ. Therefore, this study and the study by Lyall et al. are both in agreement that UKB RT and UKB Fluid IQ show relatively good stability. 

Whereas Lyall et al. found that the test-retest correlation for UKB Pairs Matching was very low (r = 0.19), here we report that this test is moderately stable (r = 0.41). The difference in the size of the test-retest correlation reported in the current study and in Lyall et al. for UKB Pairs Matching is likely to be due to differences in the length of the test-retest interval. Lyall et al.’s test retest interval was 4 years, whereas the test-retest interval in the current study was 4 weeks, which is a more conventional interval for assessing the short-term stability of a test. Given the length of time between baseline and repeat testing in Lyall et al., their test-retest correlation could reflect cognitive change over time as well as test stability. 

The discussion section has been updated to provide the reader with more detail on how the reliabilities reported in the current study compare to those reported in Lyall et al.: 

“Using the UK Biobank baseline and repeat data, Lyall et al. [3] investigated the stability of UKB Pairs Matching, UKB RT and UKB Fluid IQ. Like the current study, Lyall et al. [3] found that UKB Pairs Matching had the lowest test-retest reliability. However, the test-retest reliability for UKB Pairs Matching was substantially larger in the current study (r12 = 0.41) than was reported using UK Biobank data in Lyall et al. [3, 18], who reported the test-retest reliability of the UKB Pairs Matching test to be r12 = 0.19 [3]. The lower test-retest reliability reported in Lyall et al. [3] might be because they used a test-retest interval of over 4 years, which is much longer than the four-week test-retest interval used in the current study and therefore the test-retest correlation reported in Lyall et al. may in part reflect cognitive change over time, in addition to test stability. Despite the differences in the test-retest interval, the current study and the study by Lyall et al. [3] found very similar stability estimates for UKB RT (Lyall et al. r12 = 0.54 [3]; present study r12 = 0.55) and UKB Fluid IQ (Lyall et al. r12 = 0.65 [3]; present study r12 = 0.61), suggesting these tests do show relatively good stability.” (Page 31, lines 705-716)

21. The beginning of the ‘concurrent validity’ section is a little bit awkward, because it focuses on a very specific caveat, which should have its part in the later parts of the discussion.

Response: We have now started the ‘concurrent validity’ section of the discussion with a brief description of our findings: 

“Despite the brief and non-standard nature of the UK Biobank cognitive tests, they tended to correlate moderately-to-strongly with well-validated cognitive tests that were designed to assess the same cognitive domain or specific ability. The UK Biobank cognitive tests mostly showed modest to good concurrent validity.” (Page 26, lines 582-585)

We have moved the paragraph detailing that we are aware that most cognitive tests load strongly on a general cognitive component and therefore the reported correlations between the UK Biobank tests and the reference tests could be in part (or in whole) because they both assess general cognitive ability to the end of the ‘concurrent validity’ section:

“In addition to correlating relatively highly with the chosen reference tests, most UK Biobank cognitive tests also had positive correlations with many non-reference tests, and they loaded strongly on the general cognitive component. When we write about tests correlating because they both assess the same ‘cognitive domain’ or ‘underlying cognitive ability’ it might also be in part or in whole because they both assess general cognitive ability (g). It is an error not to acknowledge this, as Schmidt [20] discusses in detail. However, mindful of the fact that there is variance beyond g and that is accounted for at the level of cognitive domains and specific abilities [9], and the fact that readers will wish to know how the largely-undocumented UK Biobank tests relate to better-validated tests, we think the references we have made to domains and specific abilities are appropriate.” (Page 29, lines 656-664)

22. The sample demographics of the current study are not fully comparable to those of the UK Biobank. Specifically, the sample of the present study is older. This could impact the observed reliabilities and validities, and hence should be made clear in the limitations section.

Response: The age range used in the current study (40 to 80 years) is similar to the age range of the UK Biobank participants. At baseline, UK Biobank aimed to recruit participants aged 40 to 70 years old (actual age range = 37 to 73 years). These participants have, of course become older as they have been followed up. At the UK Biobank imaging visit, which uses the same cognitive assessment as was used in the current study, the age range of UK Biobank participants was 44 to 82 years. The mean age of the sample in the current study is 62.59 years (SD = 10.24), which is in between the mean age of UK Biobank participants attending the baseline visit (56.53, SD = 8.10) and the mean age of participants attending the UK Biobank imaging clinic (64.07, SD = 7.04). 

We have added more information in the methods section to make readers aware that the age range used here is similar to the range of ages of UK Biobank participants across the various data collection points: 

“The age range of 40 to 80 years was used in the current study because this is approximately the age range of UK Biobank participants across the various data collection points to date (http://biobank.ndph.ox.ac.uk/showcase/field.cgi?id=21003). The UK Biobank baseline study aimed to recruit participants aged 40 to 70 years (mean age = 56.53, SD = 8.10). These participants have become older as they have been followed up. The age range of UK Biobank participants at the UK Biobank imaging study—which uses the same cognitive assessment as in the current study—was 44 to 82 years (mean age = 62.59 years, SD = 10.24).” (Page 7, lines 140-146)

23. There are other forms of validity than concurrent validity, e.g. correlations with measures other than psychometric ones, or prediction of ‘real-world’ criteria. This should be acknowledged in the limitations section.

Response: We have updated the limitations section of the discussion to detail the fact that the current study only examined some aspects of the validity (and reliability) of the UK Biobank cognitive test, and that other forms of validity (and reliability) were not assessed here. 

“This study only examined some aspects of the validity and reliability of the UK Biobank tests. We did not examine, for example, their internal consistency or predictive validity for other ‘real-world’ outcomes.” (Page 33, line 768-770)

Reviewer 3

1. As it is stated in the manuscript, the second aim of the study was to investigate wether a component of general cognitive ability (g) was present in the correlations among the unsupervised UK Biobank tests, and wether any such g component correlated highly with a measure of general cognitive ability created using the reference tests administered by a trained tester under standardised conditions. I would like to know why information related to this aim (results and conclusions) was not included in abstract nor conclusions section

Response: We thank Reviewer 3 for their comments. We agree that it was an omission not to include the second aim of the in the abstract. We have therefore updated the abstract to additionally include information:

“Two measures of general cognitive ability were created by entering scores on the UK Biobank cognitive tests, and scores on the reference tests, respectively, into separate principal component analyses and saving scores on the first principal component. (Page 2, lines 23-25)

“The measure of general cognitive ability based on the UK Biobank cognitive tests correlated at r=0.83 (p<.001) with a measure of general cognitive ability created using the reference tests.” (Page 2, lines 28-30)

2. Was there any schedule for test-administration? If yes, was this counter-balanced? It has been shown that circadian typology can influence the results of cognitive tests depending on the time at which these tests are administered (synchronicity effect).

Response: There was a set test order in the current study and this test order was counter-balanced such that half of the participants completed the UK Biobank tests first, and half of participants completed the well-validated cognitive tests first. We counter-balanced the test order in a bid to limit the effects of fatigue on test performance. The test order for the two counter-balanced groups is reported in S1 File, Supplementary Table 2.

The testing session was quite lengthy—approximately 2.5 to 3 hours. Again to limit the effects of fatigue, all participants were given a break (with refreshments) approximately half way through the testing session. The methods section of the manuscript has been slightly updated to make readers aware that the test order was counter-balanced and that we provided participants with a break to try to alleviate the consequences of fatigue: 

“The testing session took approximately 2.5 to 3 hours to complete. To limit any effects of fatigue on test performance, the test order was counter-balanced. Individuals with even participant ID numbers completed the UK Biobank tests before completing the M-ACE and reference tests. Individuals with odd participant ID numbers completed the M-ACE and reference tests first and then completed the UK Biobank tests. The UK Biobank questionnaire was administered immediately after completing the UK Biobank cognitive assessment. Approximately half-way through the session, participants were given a short, approximately 15-20 minute break (with refreshments), again to try to limit any effects of fatigue. The test order for participants with odd and even ID numbers is shown in S1 File (Supplementary Table 2).” (Page 10-11, lines 220-228)

We did not consider circadian typology in the current study. However, the timing of the study visit was mutually agreed between the tester and the participant. To accommodate the different schedules of participants, morning, afternoon and evening appointments were available during the week, as well as at the weekend. Participants were able to select appointment times that suited them. We have updated the methods section to make readers aware that the appointment time was at a time mutually agreed by the participant and the tester: 

“Study visits took place in the Psychology Department at the University of Edinburgh at a time mutually agreed by the participant and the tester. Appointments were available in the morning, afternoon, or evening on both weekdays and weekends, to suit the participant’s schedule.” (Page 10, lines 214-216)

---

## [Decision Letter · Decision Letter 1]

30 Mar 2020

Reliability and validity of the UK Biobank cognitive tests

PONE-D-19-30264R1

Dear Dr. Fawns-Ritchie,

We are pleased to inform you that your manuscript has been judged scientifically suitable for publication and will be formally accepted for publication once it complies with all outstanding technical requirements.

The manuscript has been reevaluated by the same Reviewers who did the previous evaluation of your work. As you will see, Reviewer #1 still suggests to reject the manuscript. However, Reviewers #2 and #3 support that the manuscript is ready for publication at this point. I coincide with this later view.

With kind regards,

Angel Blanch, Ph.D.

Academic Editor

PLOS ONE

Additional Comments from the journal's editorial staff:

PLOS supports authors who wish to share their work early through deposition of manuscripts in preprint servers. This does not impact consideration of the manuscript at any PLOS journal (https://journals.plos.org/plosone/s/criteria-for-publication#loc-2).

Reviewers' comments:

Reviewer's Responses to Questions

**Comments to the Author**

1. If the authors have adequately addressed your comments raised in a previous round of review and you feel that this manuscript is now acceptable for publication, you may indicate that here to bypass the “Comments to the Author” section, enter your conflict of interest statement in the “Confidential to Editor” section, and submit your "Accept" recommendation.

Reviewer #1: (No Response)

Reviewer #2: All comments have been addressed

Reviewer #3: All comments have been addressed

2. Is the manuscript technically sound, and do the data support the conclusions?

Reviewer #1: Partly

Reviewer #2: Yes

Reviewer #3: Yes

3. Has the statistical analysis been performed appropriately and rigorously? 

Reviewer #1: I Don't Know

Reviewer #2: Yes

Reviewer #3: Yes

4. Have the authors made all data underlying the findings in their manuscript fully available?

Reviewer #1: Yes

Reviewer #2: No

Reviewer #3: Yes

5. Is the manuscript presented in an intelligible fashion and written in standard English?

Reviewer #1: Yes

Reviewer #2: Yes

Reviewer #3: Yes

6. Review Comments to the Author

Reviewer #1: I am aware of the growing use of preprints and the public debate on the issue. I strongly oppose publishing preprints and thus circulating supposedly scientific papers that possibly cloud the very issue they should shed light on. This manuscript and the comments of my esteemed colleagues reviewers prove my point lucidly, with the many errors and corrections required.

Reviewer #2: The authors have addressed my comments in a very thorough and competent manner. I have no further issues, and recommend acceptance of the manuscript.

Reviewer #3: All comments have been addressed and I consider that this manuscript has increased its quality, being suitable for its publication in PLoS ONE

7. PLOS authors have the option to publish the peer review history of their article (what does this mean?). If published, this will include your full peer review and any attached files.

Reviewer #1: Yes: D.M.Campagne

Reviewer #2: Yes: Kersten Diers

Reviewer #3: No

---

## [Editor Report · Acceptance letter]

6 Apr 2020

PONE-D-19-30264R1 

Reliability and validity of the UK Biobank cognitive tests 

Dear Dr. Fawns-Ritchie:

I am pleased to inform you that your manuscript has been deemed suitable for publication in PLOS ONE. Congratulations! Your manuscript is now with our production department. 

With kind regards,

on behalf of

Dr. Angel Blanch 

Academic Editor

PLOS ONE